# Learnware Specification via Dual Alignment

**Wei Chen** [1 2]  **Jun-Xiang Mao** [1 2 3]  **Xiaozheng Wang** [3]  **Min-Ling Zhang** [1 2]

## Abstract

The *learnware paradigm* aims to establish a learnware dock system that contains numerous learnwares, each consisting of a well-trained model and a *specification*, enabling users to reuse high-performing models for their tasks instead of training from scratch. The specification, as a unique characterization of the model's specialties, dominates the effectiveness of model reuse. Existing specification methods mainly employ distribution alignment to generate specifications. However, this approach overlooks the model's discriminative performance, hindering an adequate specialty characterization. In this paper, we claim that it is beneficial to incorporate such discriminative performance for high-quality specification generation. Accordingly, a novel specification approach named DALI, i.e., *Learnware Specification via **D**ual **ALI**gnment*, is proposed. In DALI, the characterization of the model's discriminative performance is modeled as discriminative alignment, which is considered along with distribution alignment in the specification generation process. Theoretical and empirical analyses clearly demonstrate that the proposed approach is capable of facilitating model reuse in the learnware paradigm with high-quality specification generation.

## 1  Introduction

Currently, the machine learning paradigm of step-by-step model construction from scratch, tailored to specific tasks, has achieved remarkable success. However, this success is heavily dependent on several key factors: a large volume of high-quality labeled data, significant computational re-source investment, and specialized development expertise. These factors pose substantial challenges for non-experts building high-performance models from scratch, while it is ideal if their tasks could be tackled by reusing existing well-trained models. However, issues such as data privacy, model inflexibility, and catastrophic forgetting further complicate the reuse and adaptation of models for users.

The *learnware paradigm* (Zhou, 2016) provides a systematic approach that allows users to build new machine learning solutions by leveraging existing well-established models instead of developing models from scratch. A *learnware* is a well-trained machine learning model equipped with a small-scale *specification* that describes its capabilities and specialties. This specification, serving as the unique identifier for both the learnware and its associated model, enables the model to be effectively reused by new users without requiring access to the original training data. In *the submitting stage*, developers worldwide can voluntarily submit their trained models to a *learnware dock system* (Tan et al., 2024c), and the system helps developers generate specifications corresponding to their models. In *the deploying stage*, when a user needs to address a specific task, instead of starting from scratch, she can submit her requirement to the learnware dock system. The system identifies and recommends useful learnwares, allowing the user to achieve better performance than training a model from scratch using their own data. Recently, to advance research on the learnware paradigm, the learnware dock system, *Beimingwu*, has been developed and released (Tan et al., 2024c).

Obviously, the design of the specifications is pivotal in the learnware paradigm (Zhou & Tan, 2024). Well-crafted specifications that adequately characterize models' specialties can greatly improve the accuracy of learnware identification and the effectiveness of model reuse. Existing specification methods primarily rely on distribution alignment, i.e., ensuring that the specification's distribution is similar to that of the model's training data. Although this approach partially describes the model's applicable scope from the perspective of its training data, it overlooks the model's inherent discriminative ability, resulting in an inadequate characterization of the model's specialties.

In light of the above observations, we postulate that more adequate characterization of the model's specialties can be

---

[1]School of Computer Science and Engineering, Southeast University, Nanjing, China [2]Key Lab. of Computer Network and Information Integration (Southeast University), MOE, China [3]Information Technology and Data Management Department of China Mobile Communications Group Zhejiang Co., Ltd. Correspondence to: Min-Ling Zhang <zhangml@seu.edu.cn>.

*Proceedings of the 42$^{nd}$ International Conference on Machine Learning*, Vancouver, Canada. PMLR 267, 2025.

expected if the model's discriminative performance can be appropriately considered within specification generation procedure. Accordingly, a novel specification approach named DALI, i.e., *Learnware Specification via **D**ual **ALI**gnment*, is proposed. In DALI, the characterization of the model's discriminative performance is modeled as discriminative alignment, which is carefully incorporated along with distribution alignment in the specification generation process. For discriminative alignment, our objective is to ensure that the discriminative performance of the generated specification closely approximates the model's inherent discriminative performance. For distribution alignment, we strive to guarantee that the distribution of specifications for the same class is as similar as possible to the distribution of the training data. Through this dual alignment mechanism, the DALI approach enables the generated specifications to simultaneously focus on the model from both discriminative performance and training data distribution perspectives, thereby more adequately characterizing the model's capabilities and specialties. Theoretical and empirical analyses clearly demonstrate that the proposed DALI approach is capable of facilitating model reuse in the learnware paradigm with high-quality specification generation.

## 2 Related Work

The learnware paradigm (Zhou, 2016; Zhou & Tan, 2024) offers a systematic approach to managing well-established models, enabling users to directly reuse existing models based on their specialties to address specific requirements, rather than constructing models from scratch. A learnware consists of a well-trained model and a specification describing its specialties, where the specification ensures privacy protection and unique identifiability (Lei et al., 2024; Shen & Li, 2025). As a core component of the learnware paradigm, the specification has garnered significant research attention in recent years, with substantial efforts focused on generating specifications that effectively characterize model specialties. Wu et al. 2023 proposed generating specifications with Reduced Kernel Mean Embedding (RKME) technique, ensuring that a reduced set is approximate to the task data distribution in the Reproducing Kernel Hilbert Space (RKHS). Subsequently, Zhang et al. 2021 extended this RKME specification method to handle user requirements with unseen job components. To date, RKME specification methods have achieved impressive success across various scenarios in the learnware paradigm. In the heterogeneous feature space scenarios, Tan et al. 2023 assumes access to the original training data and employs subspace learning to unify heterogeneous features into a shared feature space, thereby generating RKME specifications. To relax the strong assumption of data accessibility, Tan et al. 2024b explored the correlation of heterogeneous feature tasks, enabling the generation of RKME specifications without accessing the

original data or auxiliary data across feature spaces. To take the importance of label information into consideration, Tan et al. 2024a enhances the representation capacity of the RKME specifications by incorporating label information into the generation procedure, thereby enabling the heterogeneous learnware paradigm. Meanwhile, Guo et al. 2023 encodes label information as a part of the RKME specification to address scenarios involving heterogeneous labels. Based on the above research, the learnware dock system, *Beimingwu*, was recently released to systematically simplify the entire learnware paradigm and promote its research and practical applications (Tan et al., 2024c). Moreover, since the performance of the learnware paradigm based on RKME specifications heavily depends on the choice of kernel function, Chen et al. 2025 proposed replacing RKHS with neural embedding space to overcome this limitation, achieving specifications that align closely with class feature distributions of the training data.

Obviously, the aforementioned specification methods primarily rely on distribution alignment to generate specifications, i.e., ensuring that the specification's distribution is similar to that of the model's training data. However, neglecting the model's intrinsic discriminative performance may lead to suboptimal specifications of low-quality, thereby compromising the effectiveness of model reuse. To address this limitation, this paper proposes a novel specification approach DALI, simultaneously considering both the model's discriminative performance and the feature distribution it has mastered, in order to generate specifications that more adequately characterizations the model's specialties.

## 3 Preliminaries

The learnware paradigm consists of two distinct stages: a *submitting stage* and a *deploying stage*.

**The Submitting Stage.** In this stage, a developer can spontaneously submit her well-trained model $f$ to the learnware dock system. To better characterize the submitted model $f$, the system help the developer assign the model a specification $R$, which sketches its capabilities and specialties. Currently, the specification is primarily derived from the dataset $\mathcal{D} = \{(\mathbf{x}_i, y_i)\}_{i=1}^n$ used to train the model, serving as its unique identifier. Here, we assume that $\mathcal{D}$ comprises $K$ classes, where $\mathcal{X} = \mathbb{R}^d$ ($\mathbf{x} \in \mathcal{X}$) denotes the feature space of $\mathcal{D}$, and $\mathcal{Y} = \{1, 2, \ldots, K\}$ ($y \in \mathcal{Y}$) represents the label space of $\mathcal{D}$. The model $f$, along with its corresponding specification $R$, forms a learnware $(f, R)$, which is then stored in the learnware dock system for future use.

**The Deploying Stage.** In this stage, suppose a user has a task requirement corresponding to the dataset $\hat{\mathcal{D}} = \{(\hat{\mathbf{x}}_i, \hat{y}_i)\}_{i=1}^{\hat{n}}$. Here, we assume that $\hat{\mathcal{D}}$ comprises $\hat{K}$ classes, where $\hat{\mathcal{X}} = \mathbb{R}^{\hat{d}}$ ($\hat{\mathbf{x}} \in \hat{\mathcal{X}}$) denotes the feature space of $\hat{\mathcal{D}}$, and

$\hat{\mathcal{Y}} = \{1, 2, \ldots, \hat{K}\}$ $(\hat{y} \in \hat{\mathcal{Y}})$ represents the label space of $\hat{\mathcal{D}}$. To address this task, the system will help the user generate a specification $\hat{R}$ tailored to their requirements. The user then submit $\hat{R}$ to the learnware dock system, which will match and return potentially useful learnwares to the user based on the submitted specification $\hat{R}$. Subsequently, the user can solve the task by reusing these learnware models.

**RKME Specification.** The specification is the central part of the learnware paradigm, functioning both as a characterization of the model's capabilities and as its unique identifier. Currently, specifications primarily characterize the distributions mastered by the model while ensuring privacy protection. The RKME specification method (Tan et al., 2024c) learns the data distribution in the RKHS $\mathcal{H}_k$ through the Kernel Mean Embedding (KME) technique (Muandet et al., 2017). This approach safeguards data privacy by employing a reduced set to approximate the empirical KME. Specifically, the RKME specification is generated by optimizing the following objective:

$$\min_{\boldsymbol{\beta}, \mathbf{Z}} \left\| \frac{1}{n} \sum_{i=1}^{n} k(\mathbf{x}_i, \cdot) - \sum_{j=1}^{m} \beta_j k(\boldsymbol{z}_j, \cdot) \right\|_{\mathcal{H}_k}^2. \quad (1)$$

Here, the $m$-sized tuple $R = \{(\beta_j, \boldsymbol{z}_j)\}_{j=1}^{m}$ acts as the specification, where $m \ll n$, $\mathbf{Z} = \{\boldsymbol{z}_j\}_{j=1}^{m}$ is the reduced set, and $\boldsymbol{\beta} = \{\beta_j\}_{j=1}^{m}$ are non-negative weight coefficients. $k(\cdot, \cdot)$ is a predefined kernel function, and $\mathcal{H}_k$ is the RKHS associated with this kernel. It is evident that a major drawback of the RKME specification method is that its performance heavily relies on the predefined kernel function, which limits its applicability in different scenarios.

## 4 Methodology

In practice, the class distributions of the training data under true-labels are often entangled in the feature space. However, the corresponding well-trained model possesses discriminative ability, enabling disentanglement in the output space. If the generated specification focuses solely on the feature distribution to describe the model's capabilities, it often fails to characterize its discriminative performance, resulting in an inadequate specialty characterization. Therefore, the proposed approach considers both the model's discriminative performance and the feature distribution of the training data, providing a comprehensive characterization of the model's capabilities and specialties.

### 4.1 The Proposed of DALI Approach

Let $\{\bar{y}_i\}_{i=1}^{n}$ be the pseudo-labels generated by the model trained on the dataset $\mathcal{D} = \{(\mathbf{x}_i, y_i)\}_{i=1}^{n}$, where $\{y_i\}_{i=1}^{n}$ represents the true-labels. Generally, these two groups of labels are not identical, which leads to different latent information between the pseudo-labels and the true-labels. The

pseudo-labels convey the model's inherent discriminative performance, while the true-labels provide the class feature distribution that the model has mastered. Therefore, our goal is to ensure that the specification captures both of these types of information, thereby comprehensively characterizing the model's capabilities and specialties.

**Discriminative alignment.** According to the learnware specification mechanism, both the training data and the specification lie within a hypothesis set $\mathcal{H} \subset \{\psi : \mathcal{X} \to \mathcal{Y}\}$, where $\psi$ is the predictor (e.g., neural network). Moreover, the training data distribution in the pseudo-label space, i.e., $\bar{\mathcal{D}} = \{(\mathbf{x}_i, \bar{y}_i)\}_{i=1}^{n}$, effectively reflects the model's discriminative performance. By aligning the specification with this distribution through encoding, we can characterize the model's inherent discriminative performance. In theory, any distance metric in the distribution space can be employed to measure the difference between distributions, facilitating the corresponding approximations. However, most of these (e.g., the Kullback-Leibler divergence) are independent of the learning process of the model, making it hard to estimate from finite data and overly strict (Konstantinov & Lampert, 2019). Intuitively, a well-trained model should demonstrate performance consistent with its performance on the training data when applied to distributions that are similar to the characteristic distribution of the training data. Specifically, if a network performs well on training data distribution in the pseudo-label space but poorly on the specification, the performance discrepancy between them is significant. Conversely, if all functions in $\mathcal{H}$ exhibit similar performance on both, they are considered to be identical. From this, we can use $\mathcal{H}$-discrepancy (Konstantinov & Lampert, 2019) to measure this performance discrepancy, ensuring that the distribution of the training dataset $\bar{\mathcal{D}} = \{(\mathbf{x}_i, \bar{y}_i)\}_{i=1}^{n}$ in the pseudo-label space is aligned to that of the specification $R$, thus achieving discriminative alignment. The specific formula is as follows:

$$\mathcal{L}_{dis} = \sup_{\psi_{\boldsymbol{\theta}} \in \mathcal{H}, \boldsymbol{\theta} \sim P_{\boldsymbol{\theta}}} g(\ell(\psi_{\boldsymbol{\theta}}(\mathbf{x}), \bar{y}) - \ell(\psi_{\boldsymbol{\theta}}(\boldsymbol{z}), y_{\boldsymbol{z}})), \quad (2)$$

where $\ell$ is the cross-entropy loss function, $g(x) = \sqrt{x^2 + \alpha}$ is the smooth approximation of the absolute value operator, and $\alpha$ is a small constant. Here, the specification is denoted as $R = \{(\boldsymbol{z}_j, y_{\boldsymbol{z}_j})\}_{j=1}^{Km}$, where $Km \ll n$ with the initialization of the specification involving random sampling of $m$ data points from each class in the training data $\mathcal{D} = \{(\mathbf{x}_i, y_i)\}_{i=1}^{n}$. In addition, to avoid the inter-class dependent discriminative disparities[1] introduced by the net-

---

[1] Assume there are 3 classes of data: $\{a, b, c\}$. If the model is trained on the set $\{a, b\}$, the resulting parameters will characterize the discriminative information between $a$ and $b$, while training on $\{b, c\}$ will characterize the discriminative information between $b$ and $c$. This leads to significant variations in the model parameters related to class $b$, resulting in discrepancies in the synthesized representation of $b$ generated from different parameter sets.

work parameter $\boldsymbol{\theta}$, $\psi_{\boldsymbol{\theta}} \in \mathcal{H}$ is denoted as numerous random initialized neural networks under the distribution $P_{\boldsymbol{\theta}}$. Due to the strong representational capability of random neural networks (Saxe et al., 2011; Cao et al., 2018; Amid et al., 2022; Lee et al., 2023), which ensure the preservation of both intra-class and inter-class information when mapping data to the embedding space (Giryes et al., 2016), it can be regarded as providing a partial interpretation of the inputs. Meanwhile, a combination of numerous random neural networks offers a comprehensive interpretation. Thus, numerous random neural networks can effectively encode the discriminative performance in the pseudo-label space and, through $\mathcal{H}$-discrepancy, align the specification with it for discriminative alignment, thereby characterizing the well-trained model's discriminative performance.

**Distribution alignment.** While characterizing the model's discriminative performance, we also desire the specification to capture the feature distribution provided by the true-label space. This provides a more comprehensive characterization of the model's capabilities. To achieve this, in numerous random neural embedding spaces, we use the Maximum Mean Discrepancy (MMD) to compute the unbiased estimate of the class expectation between the training data $\mathcal{D} = \{(\mathbf{x}_i, y_i)\}_{i=1}^n$ in the true-label space and the specification $R$, ensuring that their class feature distributions are aligned. The formula is expressed as:

$$
\mathcal{L}_{MMD} = \mathbb{E}_{\boldsymbol{\vartheta} \sim P_{\boldsymbol{\vartheta}}} \sum_{k=1}^{K} \left\| \frac{1}{|\mathcal{D}_k|} \sum_{(\mathbf{x},y) \in \mathcal{D}_k} \psi_{\boldsymbol{\vartheta}}(\mathbf{x}) - \frac{1}{|R_k|} \sum_{(\boldsymbol{z}, y_{\boldsymbol{z}}) \in R_k} \psi_{\boldsymbol{\vartheta}}(\boldsymbol{z}) \right\|^2, \tag{3}
$$

where $\mathcal{D}_k$ and $R_k$ represent the subsets of the training data and the specification corresponding to the $k$-th class, respectively. $\psi_{\boldsymbol{\vartheta}} : \mathbb{R}^d \to \mathbb{R}^{d'}$ is the embedding neural network, where $d' \ll d$ and $\boldsymbol{\vartheta} \sim P_{\boldsymbol{\vartheta}}$ represents the randomly initialized parameters under the distribution $P_{\boldsymbol{\vartheta}}$ of the embedding network parameter.

**Overall.** During the overall specification generation process, by integrating discriminative alignment and distribution alignment, the objective function of the proposed DALI approach can be expressed as:

$$
\min_{\boldsymbol{z}} \mathcal{L}_{dis} + \mathcal{L}_{MMD}. \tag{4}
$$

It is worth noting that the specification $R = \{(\boldsymbol{z}_j, y_{\boldsymbol{z}_j})\}_{j=1}^{Km}$ characterizes the well-trained model's discriminative performance and mastered feature distribution by simultaneously aligning the discriminative performance in the pseudo-label space and the class feature distributions in the true-label space across numerous random neural networks, thereby

achieving an adequate characterization of the model's capabilities and specialties. The overall procedure of the DALI approach is outlined in Algorithm 1 of Appendix A.

The following detailed propositions provide a theoretical analysis of the expected loss bound of the proposed DALI approach, as well as its privacy protection. For simplicity, we consider only one class and omit the label vectors, while also providing some assumptions.

**Assumption 4.1.** Let the random neural network $\psi_{\boldsymbol{\vartheta}} : \mathbb{R}^d \to \mathbb{R}^{d'}$ be considered nonlinear, where $\boldsymbol{\vartheta}_l$ represents the weight parameter of the $l$-th layer and the corresponding activation function is denoted as $\rho_l$. Except for the final layer, which is a linear classification layer, the other layers of the network $\psi_{\boldsymbol{\theta}} : \mathbb{R}^d \to \mathbb{R}^K$ are similar to those of $\psi_{\boldsymbol{\vartheta}}$.

**Assumption 4.2.** We assume here at least one specification $R^* = \{\boldsymbol{z}_1^*, \cdots, \boldsymbol{z}_{|R|}^*\}$ that minimizes Eq.(4).

For further simplification in analysis, we separately perform the upper bound analysis for $\mathcal{L}_{dis}$ and $\mathcal{L}_{MMD}$. The details of upper bound analysis are as follows:

**Proposition 4.3** (Upper bound for $\mathcal{L}_{dis}$). *According to the theorem 1 of (Mohri & Muñoz Medina, 2012), we assume that the loss function $\ell$ is bounded by $M$, where $M$ lies between 0 and 1. Let a hypothesis set $\mathcal{H} \subset \{\psi : \mathcal{X} \to \mathcal{Y}\}$. For any $\delta > 0$, with probability at least $1 - \delta$ over the data, the following holds for $\psi_{\theta}$:*

$$
\begin{aligned}
\mathcal{L}_{dis} &= |\ell(\psi_\theta(\mathbf{x}), \bar{y}) - \ell(\psi_\theta(\boldsymbol{z}), y_{\boldsymbol{z}})| \\
&\leq 2qM^{q-1}\Re(\psi) + disc_{\mathcal{H}}(\mathcal{D}, R) + M\sqrt{\frac{\log \frac{1}{\delta}}{2}}.
\end{aligned} \tag{5}
$$

*where*

$$
\Re(\psi) = \mathbb{E}_{\boldsymbol{\sigma}} \left[ \sup_{\psi_\theta \in \psi} \sigma \ell(\psi_\theta(\mathbf{x}), \bar{y}) \right],
$$

*and $\sigma$ are independent Rademacher random variables. In addition, the $\psi_\theta$ is the random initialized neural network under distribution $P_{\boldsymbol{\theta}}$, and $\ell$ is the $\ell_q$ loss function.*

**Proposition 4.4** (Upper bound for $\mathcal{L}_{MMD}$). *Let $\mathbf{x} \in \{\mathbf{x}_i | (\mathbf{x}_i, y_i) \in \mathcal{D}\}$ and $\boldsymbol{z} \in \{\boldsymbol{z}_i | (\boldsymbol{z}_i, y_{\boldsymbol{z}_i}) \in R\}$ be considered Borel probability measures in the data topological space $\mathcal{X}$. According to (Golowich et al., 2018), the empirical Rademacher complexity of $\psi$ for dataset $\mathcal{D}$ is defined by:*

$$
\Re_{\mathcal{D}}(\psi) \leq \frac{B(\sqrt{2\log(2)l} + 1) \prod_{j=1}^{l} M_F(j)}{\sqrt{|\mathcal{D}|}}, \tag{6}
$$

*where $\psi$ is the embedding network of depth $l$ over the topological space $\mathcal{X}$, and each layer network parameter matrix $\boldsymbol{\vartheta}_j$ has a Frobenius norm bounded by $M_F(j)$. The activation function is a 1-Lipschitz and positive-homogeneous*

*activation function (such as the ReLU ), and the input satisfies $\|\mathbf{x}\| \leq B$. If $\mathcal{D} \sim R$, then, with probability at least $1 - \delta$, and for any arbitrary small $\delta > 0$, the empirical MMD with neural embedding is bounded by:*

$$\mathcal{L}_{MMD}(\mathcal{D}, R)$$

$$\leq 2\Re_{\mathcal{D}} + 2\Re_R + \Re_{\mathcal{D}}\Re_R \sqrt{\frac{(|\mathcal{D}| + |R|)\log\frac{1}{\delta}}{2|\mathcal{D}||R|}}. \quad (7)$$

The detailed relevant proof can be found in Appendix B. By combining the upper bound analysis of Proposition 4.3 and 4.4, in the DALI approach, we bound the gap between the empirical loss of the training data in neural networks and the expected loss of the specification. This bound is influenced by performance discrepancy, network parameter norms, and network depth. However, by leveraging numerous random neural networks, we can mitigate the impact of these factors on the discrepancy, enabling the specification to comprehensively characterize the model's discriminative performance and mastered feature distribution.

Based on Assumption 4.1 and 4.2, we can further prove the minimization of the proposed DALI approach's objective function loss, while ensuring privacy protection.

**Proposition 4.5** (Minimizer of Eq.(4)). *In the DALI approach, the specification $R$ is initialized by randomly selecting $|R|$ samples from $\mathcal{D}$, i.e., $\forall \boldsymbol{z} \in \{\boldsymbol{z}_i | (\boldsymbol{z}_i, y_{\boldsymbol{z}_i}) \in R\}$, $\exists \mathbf{x} \in \{\mathbf{x}_i | (\mathbf{x}_i, y_i) \in \mathcal{D}\}, \boldsymbol{z} = \mathbf{x}$. Then, the specification $R^*$ is optimized using Eq.(4) to ensure that the model performance of $R^*$ and $\mathcal{D}$ are consistent under the hypothesis set $\mathcal{H}$, and that the barycenters of $R^*$ coincides with the barycenters of $\mathcal{D}$. Specifically, it can be expressed as:*

$$\mathcal{L}(\boldsymbol{\theta}_{l+1}(\rho_l \cdots (\rho_1(\boldsymbol{\theta}_1 \cdot \mathbf{x}))), \bar{y}) -$$
$$\mathcal{L}(\boldsymbol{\theta}_{l+1}(\rho_l \cdots (\rho_1(\boldsymbol{\theta}_1 \cdot \boldsymbol{z}^*))), y_{\boldsymbol{z}}) +$$
$$\frac{1}{|\mathcal{D}|} \sum_{n=1}^{|\mathcal{D}|} \rho_l(\boldsymbol{\vartheta}_l \cdots (\rho_1(\boldsymbol{\vartheta}_1 \cdot \mathbf{x}_n))) - \quad (8)$$
$$\frac{1}{|R|} \sum_{m=1}^{|R|} \rho_l(\boldsymbol{\vartheta}_l \cdots (\rho_1(\boldsymbol{\vartheta}_1 \cdot \boldsymbol{z}_m^*))) \to 0.$$

The relevant proof can be found in Appendix C. This indicates that the proposed DALI method can optimize the discriminative alignment and distribution alignment between the specification and the training dataset.

**Proposition 4.6** (Private protection). *Based on to Proposition 4.5, Assumption 4.2 and the specification $R$ is initialized by randomly selecting $|R|$ samples from $\mathcal{D}$, i.e., $\forall \boldsymbol{z} \in \{\boldsymbol{z}_i | (\boldsymbol{z}_i, y_{\boldsymbol{z}_i}) \in R\}, \exists \mathbf{x} \in \{\mathbf{x}_i | (\mathbf{x}_i, y_i) \in \mathcal{D}\}, \boldsymbol{z} = \mathbf{x}$, we can derive the conclusion:*

$$\boldsymbol{z}_i^* = \mathbf{x}_j + \ell(\psi_{\boldsymbol{\theta}}(\mathbf{x}), \bar{y}) - \ell(\psi_{\boldsymbol{\theta}}(\boldsymbol{z}), y_{\boldsymbol{z}})$$
$$+ \frac{1}{|\mathcal{D}_k|} \sum_{j \in \mathcal{D}_k} \psi_{\boldsymbol{\vartheta}}(\mathbf{x}_j) - \frac{1}{|R_k|} \sum_{j \in R_k} \psi_{\boldsymbol{\vartheta}}(\boldsymbol{z}_j). \quad (9)$$

The relevant proof can be seen in Appendix D. Evidently, in the DALI approach, the discrepancy between the initial specification and the pseudo-barycenter of the training data decreases as the specification size increases. Conversely, the discrepancy becomes more pronounced when the specification size is smaller. This offers a compelling rationale for why a larger specification size could lead to data leakage. However, in the learnware paradigm, the specification size remains significantly smaller than the training data size, thereby ensuring the privacy protection of DALI approach.

### 4.2 The Submitting Stage

In this stage, a developer spontaneously submits her task and well-trained model to the learnware dock system to generate a specification. This specification, combined with the model, constitutes the learnware, which is subsequently stored within the system. Suppose the developer can voluntarily submit their well-trained model $f : \mathcal{X} \to \mathcal{Y}$ along with a local private training dataset $D = \{(\mathbf{x}_i, y_i)\}_{i=1}^n$. Since the DALI approach requires additional discriminative performance from the well-trained model, the outputs of the model on the training data are used as pseudo-labels. This is specifically represented as follows:

$$\bar{y}_i = f(\mathbf{x}_i), \ i = 1, 2, \ldots, n, \ \bar{y} \in \mathcal{Y} \quad (10)$$

This label setup ensures that the generated pseudo-labels encapsulate the discriminative performance of the will-trained model. Then, the system processes the dataset $\{(\mathbf{x}_i, y_i, \bar{y}_i)\}_{i=1}^n$, which includes both true-labels and the pseudo-labels, using Eq.(4) to generate a specification $R = \{(\boldsymbol{z}_j, y_{\boldsymbol{z}_j})\}_{j=1}^{Km}$ that completely characterizes the model's specialties. This specification $R$ is subsequently combined with the well-trained model $f$ to form the learnware $\{f, R\}$, which is stored within the system. The detailed procedure is provided in Algorithm 2 of Appendix A.

### 4.3 The Deploying Stage

At this stage, users search for useful learnwares in the learnware dock system based on their requirements, reusing either a single well-established model or a combination of models to fulfill their requirements. According to the DALI approach in Eq.(4), the requirement dataset $\hat{\mathcal{D}} = \{(\hat{\mathbf{x}}_i, \hat{y}_i)\}_{i=1}^{\hat{n}}$ submitted by the user is first processed to generate a specification $\hat{R} = \{(\hat{\boldsymbol{z}}_j, y_{\hat{\boldsymbol{z}}_j})\}_{j=1}^{\hat{K}\hat{m}}$ representing the requirement. In this process, pseudo-labels are replaced with true-labels, ensuring that the specification generated under discriminative alignment and distribution alignment maintains feature distribution similarity while preserving discriminability among class distributions. Here, the learnware dock system contains $c$ learnwares. Then, the system searches for matching existing learnware specifications through the requirement specification, identifying useful learnwares to

meet the user requirement. This process is expressed as:

$$\min_{\mathbf{W}} \left\| \frac{1}{|\hat{R}|} \sum_{(\hat{z}, \hat{y}_{\hat{z}}) \in \hat{R}} \hat{z} - \sum_{i=1}^{c} \sum_{j=1}^{K_i} w_{i,j} \frac{1}{|R_{i,j}|} \sum_{(z, y_z) \in R_{i,j}} z \right\|^2. \tag{11}$$

Here, $R_{i,j}$ represents the $j$-th class specification in the $i$-th learnware. The relationship between the existing learnware specifications and requirement specification is denoted by $\mathbf{W} = [w_{1,1}, \dots, w_{c,K_c}]^\top$, where $\sum_{i=1}^{c} \sum_{j=1}^{K_i} w_{i,j} = 1$, and $\sum_{j=1}^{K_i} w_{i,j}$ represents the usefulness of the $i$-th learnware in addressing the user requirement. $\mathbf{W}$ can be obtained by optimizing the quadratic programming problem shown in Eq.(11) using any off-the-shelf solvers.

After this process, we can obtain $c'$ candidate useful learnwares by $\sum_{j=1}^{K_i} w_{i,j}$ (ranked from largest to smallest), where $c' = \hat{K}$. Then, due to the potential heterogeneity of label spaces, to enhance learnware model reuse, we utilize *Cosine* similarity to quantify the differences between the classes in the requirement specification and those in the candidate learnware specifications. This similarity establishes the relationship between the learnware models and the requirement class data, and is expressed as:

$$\text{Similarity}_{Cosine}\left(\frac{1}{|\hat{R}_{\hat{i}}|} \sum_{(\hat{z}, \hat{y}_{\hat{z}}) \in \hat{R}_{\hat{i}}} \hat{z}, \frac{1}{|R_{i,j}|} \sum_{(z, y_z) \in R_{i,j}} z\right) \tag{12}$$

where $\hat{i} \in [\hat{K}]$, $i \in [c']$ and $j \in [K_i]$. Subsequently, a bipartite graph is constructed based on the *Cosine* similarity between the classes of the requirement specification and the classes of the candidate learnware specifications, where the edge weights represent the similarity. The two node sets correspond to the classes of the requirement specification and the learnware specifications, respectively. Through the Hungarian algorithm (Kuhn, 1955), we can obtain the maximum matching for each bipartite graph. This enables the system to identify the most similar learnware based on the maximum class similarity, thereby allowing the corresponding requirement class data to reuse the associated learnware's well-established model $f$. The procedure of this stage is detailed in Algorithm 3 of Appendix A.

## 5 Experiments

To demonstrate the superior specification quality of the proposed DALI approach in the learnware paradigm, we conduct comparative experiments under different (homogeneous or heterogeneous) label space settings and mixed task settings. Furthermore, we perform additional tests to verify the privacy protection and provide a visualization of the specification. Additionally, an ablation study is carried out to analyze each alignment component of the DALI approach.

### 5.1 Experimental Setup

**Dataset.** We evaluate the learnware paradigm by extracting and reconstructing datasets from DomainNet (Peng et al., 2019) and NICO (He et al., 2021). These two image datasets are commonly used to assess the effectiveness of model reuse. Specifically, both datasets contain domains with overlapping classes. In the NICO dataset, we selected 6 different domains: [autumn, dim, grass, outdoor, rock, water]. The DomainNet dataset includes 5 domains: [clipart, infograph, painting, quickdraw, real]. Such hierarchical structures are well-suited for the learnware paradigm. Here, we extract 4 label spaces from the overlapping classes of 11 domains across the two datasets, which facilitate the subsequent experimental design of developer tasks and user requirements according to different scenarios in the learnware paradigm. The specific label spaces are as follows:

- *Label space A*: ["flower", "horse", "cow", "rabbit", "tiger", "bird", "bus", "sailboat", "train", "helicopter"]

- *Label space B*: ["butterfly", "owl", "bird", "giraffe", "frog", "squirrel", "train", "tent", "truck", "umbrella"]

- *Label space C*: ["flower", "horse", "cow", "rabbit", "tiger"]

- *Label space D*: ["bird", "giraffe", "frog", "squirrel", "tiger"]

**Implementation details.** In this experiment, we use RKME, RKME-W (Guo et al., 2023), and LANE specification methods as baselines, with parameters set to their optimal choices. To ensure the fairness of the comparative experiments, the global feature extractor and the well-established models included in the learnware dock system are derived from *DenseNet201* (Huang et al., 2017) and *ResNet18* (He et al., 2016). The random neural network $\psi$ in the DALI approach is set to *ConvNetBN* (Rawat & Wang, 2017). The detailed implementation can be found in Appendix E.

### 5.2 Different Label Space Settings

The proposed DALI approach incorporates label information, while in the learnware paradigm, both homogeneous and heterogeneous label space scenarios exist between the specification label space and the user requirement label space. Therefore, we use different label space settings to validate the proposed approach. In the homogeneous label space setting, we have 11 domains $\times$ 2 label spaces (*A* and *B*) = 22 developer tasks. Each model trained on these tasks is treated as a learnware's well-trained model, and the corresponding specifications are used together to construct the learnware dock system. Moreover, the data not included in the 22 tasks is treated as the 22 user requirements. For each user requirement, the system contains 22 candidate

Table 1. The result of $Pre@k$ (%) metric in the useful learnwares identification under different label space settings.

| LABEL SETTING | METHODS | SIZE | PRE@1 | PRE@2 |
|---|---|---|---|---|
| HOMOGENEOUS LABEL SPACE | RKME | 20 | 50.00 | 77.27 |
| | | 100 | 40.91 | 63.64 |
| | RKME-W | 20 | 100.00 | 100.00 |
| | | 100 | 100.00 | 100.00 |
| | LANE | K×5 | 90.91 | 100.00 |
| | | K×10 | 95.94 | 100.00 |
| | DALI | K×5 | 100.00 | 100.00 |
| | | K×10 | 100.00 | 100.00 |
| HETEROGENEOUS LABEL SPACE | RKME | 20 | 45.46 | 54.55 |
| | | 100 | 59.10 | 63.63 |
| | RKME-W | 20 | 63.64 | 68.18 |
| | | 100 | 63.64 | 68.18 |
| | LANE | K×5 | 63.64 | 68.18 |
| | | K×10 | 63.64 | 63.64 |
| | DALI | K×5 | 68.18 | 72.73 |
| | | K×10 | 63.64 | 63.64 |

learnwares. In the heterogeneous label space setting, we treat data from the 11 domains and label spaces *C/D* as the user requirements, and the data from the 11 domains and label spaces *A/B* as the developer tasks. In this setting, no learnware in the system shares the same label space as the user requirement. Here, we use $Pre@k$ (Guo et al., 2023) as the evaluation metric for this experiment to assess the performance of the learnware dock systems, constructed with different methods under two label space settings, in identifying useful learnwares based on user requirements. The $Pre@k$ metric is defined as:

$$Pre@k = \frac{1}{\mathcal{T}} \sum_{t=1}^{\mathcal{T}} \mathbb{I}(\pi_t^{f_{best}} \leq k), \quad (13)$$

where $\pi^f$ is ranked from smallest to largest based on the similarity measure between the learnware specifications and the requirement specification, while also considering the performance of each learnware model $f$ on requirement (ranked from largest to smallest). $\mathcal{T}$ represents the number of requirements. The specific experimental results are shown in Table 1. The results show that the proposed DALI approach achieves precise identification of useful learnwares in homogeneous label space and maintains high identification accuracy even in the more challenging heterogeneous environment. Furthermore, under two different specification sizes, the proposed approach outperforms or matches other methods in both label space scenarios.

### 5.3 Mixed Task Setting

One important purpose of learnware paradigm is to enable well-trained models in the learnware dock system to be used "beyond the capabilities of any single model". Therefore, we use the mixed task setting to validate the proposed approach. In this setting, we have 11 domains × 2 label spaces

Table 2. The result of *superclass accuracy* (%) and *quality* (%) in the mixed task setting of learnware paradigm. Noting that *Label* refers to the real label of the specification.

| METHODS | SIZE | ACC | QUALITY | LABEL? |
|---|---|---|---|---|
| RKME | 20 | 63.64 | - | × |
| | 100 | 90.91 | - | × |
| RKME-W | 20 | 100.00 | - | × |
| | 100 | 100.00 | - | × |
| LANE | K×5 | 95.94 | 53.19 | √ |
| | K×10 | 100.00 | 90.23 | √ |
| DALI | K×5 | 100.00 | 75.68 | √ |
| | K×10 | 100.00 | 94.05 | √ |

(A and B) = 22 developer tasks as superclasses, i.e., 22 superclasses; and label space $A \cup B \times 11$ domains = 11 user requirement tasks. Thus, each requirement task contains two superclasses, i.e., each requirement task requires two corresponding learnware to be solved. This ensures that no single learnware model in the system can solve the requirement independently, but rather, a combination of models is required to address the requirement. Furthermore, in order to evaluate the performance of the learnware dock system constructed by each specification method, it is implemented here by evaluating the accuracy of the system in identifying useful learnware, i.e., the superclass accuracy. Meanwhile, our proposed method is to generate a class specification. In order to verify the inter-class discriminability of the class specification, it is evaluated here by accessing the class accuracy of the corresponding specifications through the developer's pre-trained model test, i.e., *quality*. Table 2 shows that the proposed approach outperforms or matches other methods in both metrics under the mixed task setting. Moreover, the DALI approach outperforms the LANE method in *quality* metrics, indicating that the DALI provides a superior specification quality compared to methods that only capture the mastered feature distribution of the model. Note that the specifications of the related RKME methods do not have labels, so the *quality* assessment cannot be performed.

### 5.4 Further Analysis

**Privacy protection analysis.** Privacy protection in the specification has always been a key concern in the learnware paradigm. Here, we assess the privacy protection performance of the proposed approach using the Binary Classifier based Membership Inference Attack (MIA) in the White-Box Setting method (Hu et al., 2022). In this method, the

Table 3. The prediction correctness (%) result of the Membership Inference Attack in the White-Box Setting method.

| METHODS | ACCURACY | ROC | PRECISION |
|---|---|---|---|
| RANDOM | 90.55±0.16 | 90.55±0.16 | 84.48±0.30 |
| RKME | 66.13±0.81 | 66.13±0.81 | 60.32±0.44 |
| LANE | 64.00±0.52 | 64.00±0.52 | 58.55±0.27 |
| DALI | 64.57±0.49 | 64.57±0.49 | 58.92±0.24 |

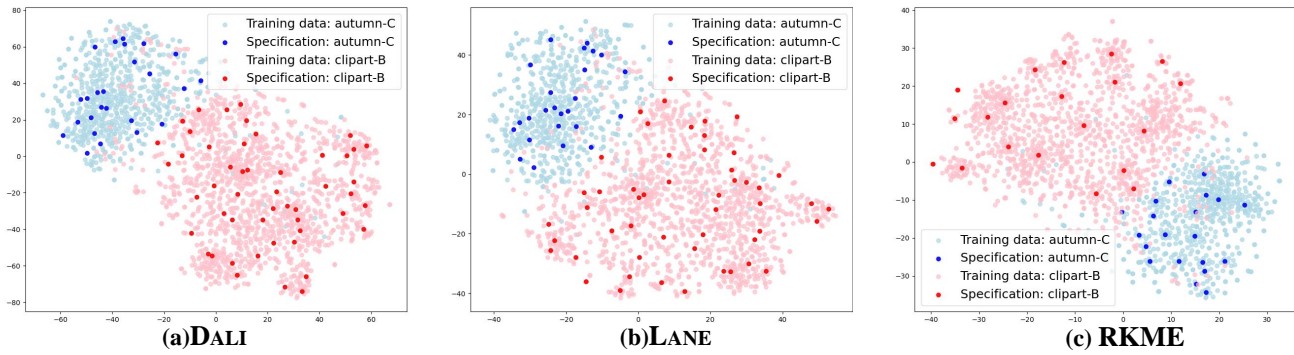

*Figure 1.* Visualization of the training data and specification from the task perspective.

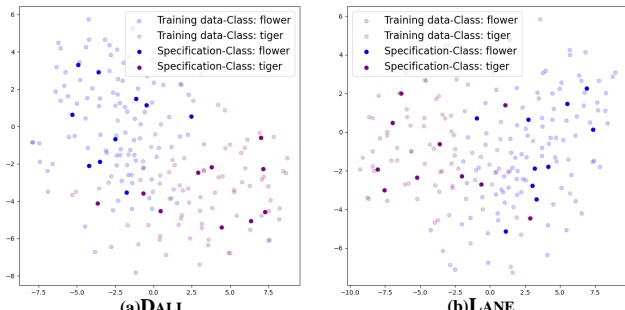

*Figure 2.* Visualization of the training data and specification from the task class perspective.

target model is obtained using the training data as member data, and the feature representations of non-member specification and member data are recorded in the target model. Then, a binary classification attack model is trained using the recorded feature representations. Finally, the attack model predicts whether the specification corresponds to member or non-member, and the evaluation is performed using three classification metrics: *Accuracy*, *ROC* and *Precision*. Note that in this experiment, the specification size is set to 100, and the random method involves randomly selecting 100 samples from the training data as no-member data. Table 3 shows that, except for the random method, other specification methods exhibit privacy protection capability.

**Visualization analysis.** To further highlight the superior specification quality of the DALI approach, we use the t-SNE method to visually represent the training data from different domains and the corresponding specifications under various methods. Figure 1 shows that all three methods effectively enable the specifications to capture the feature distributions of the tasks that the model is proficient in. Moreover, we provide a visualization of the class data and class specifications from the task, as shown in Figure 2. It is apparent that the DALI approach's class specification is more discriminative than the class specification of the LANE method. This indicates that the specification, by capturing both the model's discriminative performance and the mastered feature distribution, achieves an adequate characterization of the model's capabilities and specialties.

*Table 4.* Ablation study results of DALI approach for identifying useful learnwares by $Pre@k$ metric under different label space settings.

| Label Setting | $\mathcal{L}_{dis}$ | $\mathcal{L}_{MMD}$ | Pre@1 | Pre@2 |
|---|---|---|---|---|
| Homogeneous Label Space | ✓ | | 59.09 | 90.91 |
| | | ✓ | 90.91 | 100.00 |
| | ✓ | ✓ | 100.00 | 100.00 |
| Heterogeneous Label Space | ✓ | | 59.09 | 63.64 |
| | | ✓ | 63.64 | 68.18 |
| | ✓ | ✓ | 68.18 | 72.73 |

**Analysis of the ablation study.** The objective function of the proposed DALI approach consists of two components: one that is discriminative alignment, denoted as $\mathcal{L}_{dis}$, and another that is distribution alignment, denoted as $\mathcal{L}_{MMD}$. Here, we validate and analyze the effectiveness of these two alignment components through an ablation study. Table 4 presents the ablation study results of the $Pre@k$ evaluation for identifying useful learnwares under different label space settings. The results show that the specification generated by each individual component achieves good accuracy in identifying useful learnwares, while the combination of both components yields the greatest performance improvement.

## 6 Conclusion

In summary, this paper introduces the DALI approach, which achieves more adequate characterization of model's capabilities and specialties through discriminative alignment and distribution alignment. Discriminative alignment is accomplished by modeling the model's intrinsic discriminative performance to ensure that the generated specification's discriminative performance aligns with its. Simultaneously, distribution alignment is realized by ensuring the specification's class feature distribution closely resembles that of the training data. This enables the constructed learnware dock system to handle various label space scenarios beyond mixed tasks. Theoretical and experimental analyses demonstrate that the proposed approach achieves superior specification quality in the learnware paradigm.

## Impact Statement

This paper presents work whose goal is to advance the field of Machine Learning. There are many potential societal consequences of our work, none which we feel must be specifically highlighted here.

## Acknowlegement

The authors wish to thank the anonymous reviewers for their helpful comments and suggestions. This work was supported by the National Science Foundation of China (62225602), the Postgraduate Research & Practice Innovation Program of Jiangsu Province (KYCX25_0484), and the Big Data Computing Center of Southeast University.

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

## A Algorithm details

The detailed procedure of the proposed DALI approach is presented in Algorithm 1. First of all, the specification is initialized through random sampling from the developer dataset $\mathcal{D}$ (Step 1). Then, among numerous random neural networks, the model's discriminative performance and the feature distributions of training dataset are encoded, allowing the specification to align both discriminatively and distributively during the optimization process (Step 2-7). Finally, the generated specification is combined with the well-established model to create the learnware.

The proposed approach, as an integral part of the learnware paradigm, will be applied in this framework. The learnware paradigm is primarily divided into two stages: the submitting stage and the deploying stage. The detailed procedures for the submitting and deploying stages based on the DALI approach are outlined in Algorithm 2 and Algorithm 3.

---

**Algorithm 1** The DALI approach

---

**Input:** Dataset $\mathcal{D} = \{(\mathbf{x}_i, y_i)\}_{i=1}^n$ with pesudo-lables $\{\bar{y}_i\}_{i=1}^n$.
**Parameter:** Neural network $\psi_{\boldsymbol{\theta}}$ parameterized with $\boldsymbol{\theta}$, random probability distribution over parameters $P_{\boldsymbol{\theta}}$, neural embedding network $\psi_{\boldsymbol{\vartheta}}$ parameterized with $\boldsymbol{\vartheta}$, random probability distribution over parameters $P_{\boldsymbol{\vartheta}}$, iteration $T$, learning rate $\eta$.

1: Randomly initialize the specification $R$ from $\mathcal{D}$, with each class having a size of $m$, and the specification class label $y_z$ is consistent with the true label $y$ of the sampled data class;
2: **for** $i = 1$ **to** $T$ **do**
3:     Sample $\boldsymbol{\theta} \sim P_{\boldsymbol{\theta}}$;
4:     Sample $\boldsymbol{\vartheta} \sim P_{\boldsymbol{\vartheta}}$;
5:     Compute the objective function DALI approach of Eq.(4);
6:     Update $\boldsymbol{z}$ by the Adam optimizer;
7: **end for**

**Output:** Specifiction $R = \{(\boldsymbol{z}_j, y_{\boldsymbol{z}_j})\}_{j=1}^{Km}$.

---

**Algorithm 2** The Submitting stage

---

**Input:** Developer training dataset $\mathcal{D} = \{\mathbf{x}_i, y_i\}_{i=1}^n$, well-trained model $f$, and learnware dock system.

1: Obtain the pseudo-labels $\{\bar{y}_i\}_{i=1}^n$ by Eq.(10);
2: Obtain the specification $R$ by Alg.1;
3: Learnware $\{f, R\}$ is stored in the learnware dock system.

---

**Algorithm 3** The Deploying stage

---

**Input:** User requirement dataset $\hat{\mathcal{D}} = \{(\hat{\mathbf{x}}_i, \hat{y}_i)\}_{i=1}^{\hat{n}}$ and learnware dock system.

1: Load the learnware dock system;
2: Obtain the specification $\hat{R}$ by Alg.1, where the pseudo-labels are substituted with true labels;
3: Compute $\mathbf{W}$ by Eq.(11);
4: Select $c'$ candidate learnware by $\mathbf{W}$;
5: Compute the *similarity* between each class in the specification $\hat{R}$ and each class in the $c'$ candidate learnware specification by Eq.(12) to construct a bipartite graph;
6: Run the Hungarian algorithm to find the maximum matching for each specification $\hat{R}$ class;
7: Reuse the corresponding learnware model $f$ for the corresponding requirement $\hat{D}$ class by maximum matching.

---

## B Proof of the upper bound of DALI specification.

For further simplification in analysis, we transform the upper bound analysis of the DALI approach into separate upper bound analyses for $\mathcal{L}_{dis}$ and $\mathcal{L}_{MMD}$, where the formulas of $\mathcal{L}_{dis}$ and $\mathcal{L}_{MMD}$ are given Eq.(2) and Eq.(3).

## B.1 The upper bound of $\mathcal{L}_{dis}$

According to the theorem 1 of (Mohri & Muñoz Medina, 2012), let $\Phi$ be the function defined over any sample $\mathcal{D}$ and $R$ in the $(\mathcal{X} \times \mathcal{Y})$ by

$$\Phi(R) = \sup_{\psi_{\boldsymbol{\theta}} \in \psi} \mathbb{E}_{(\mathbf{x},y) \sim \mathcal{D}}[\ell(\psi_{\boldsymbol{\theta}}(\mathbf{x}), \bar{y})] - \ell((\psi_{\boldsymbol{\theta}}(\boldsymbol{z}), y_{\boldsymbol{z}})). \tag{14}$$

where $\mathcal{H} \subset \{\psi : \mathcal{X} \to \mathcal{Y}\}$. In addition, $\psi_{\boldsymbol{\theta}}$ is denoted as numerous random initialized neural networks under the distribution $P_{\boldsymbol{\theta}}$. Now, we assume that $(\boldsymbol{z}, y_{\boldsymbol{z}}) \in R$ and $(\boldsymbol{z}', y_{\boldsymbol{z}}') \in R'$ are two sample sets differing by one labeled point. We have:

$$\Phi(R) - \Phi(R') \leq \sup_{\psi_{\boldsymbol{\theta}} \in \psi} [\ell(\psi_{\boldsymbol{\theta}}(\boldsymbol{z}), y_{\boldsymbol{z}}) - \ell(\psi_{\boldsymbol{\theta}}(\boldsymbol{z}'), y_{\boldsymbol{z}}')] \leq M, \tag{15}$$

where $M$ is a constant and $0 \leq M \leq 1$. Thus, by the McDiarmid's inequality theorem (McDiarmid, 1989), the following holds:

$$\Pr_{R \sim \mathcal{D}}[\Phi(R) - \mathbb{E}_{R \sim \mathcal{D}}[\Phi(R)] > \epsilon] \leq \exp(-\frac{2\epsilon^2}{M}) \tag{16}$$

Then, the bound of $\mathbb{E}_{R \sim \mathcal{D}}[\Phi(R)]$ is given as:

$$\begin{aligned}
&\mathbb{E}_{R \sim \mathcal{D}}[\Phi(R)] \\
=&\mathbb{E}_{R \sim \mathcal{D}}\left[\sup_{\psi_{\boldsymbol{\theta}} \in \psi} \mathbb{E}_{(\mathbf{x},\bar{y}) \sim \mathcal{D}}[\ell(\psi_{\boldsymbol{\theta}}(\mathbf{x}), \bar{y})] - \ell((\psi_{\boldsymbol{\theta}}(\boldsymbol{z}), y_{\boldsymbol{z}})\right] \\
=&\mathbb{E}\left[\sup_{\psi_{\boldsymbol{\theta}} \in \psi} \mathbb{E}_{(\mathbf{x},\bar{y}) \sim \mathcal{D}}[\ell(\psi_{\boldsymbol{\theta}}(\mathbf{x}), \bar{y})] - \mathbb{E}_{(\boldsymbol{z},y_{\boldsymbol{z}}) \sim R}[\ell(\psi_{\boldsymbol{\theta}}(\boldsymbol{z}), y_{\boldsymbol{z}})] + \mathbb{E}_{(\boldsymbol{z},y_{\boldsymbol{z}}) \sim R}[\ell(\psi_{\boldsymbol{\theta}}(\boldsymbol{z}), y_{\boldsymbol{z}})] - \ell((\psi_{\boldsymbol{\theta}}(\boldsymbol{z}), y_{\boldsymbol{z}})\right] \\
\leq&\mathbb{E}\left[\sup_{\psi_{\boldsymbol{\theta}} \in \psi} \mathbb{E}_{(\mathbf{x},\bar{y}) \sim \mathcal{D}}[\ell(\psi_{\boldsymbol{\theta}}(\mathbf{x}), \bar{y})] - \mathbb{E}_{(\boldsymbol{z},y_{\boldsymbol{z}}) \sim R}[\ell(\psi_{\boldsymbol{\theta}}(\boldsymbol{z}), y_{\boldsymbol{z}})]\right] + \mathbb{E}\left[\sup_{\psi_{\boldsymbol{\theta}} \in \psi} \mathbb{E}_{(\boldsymbol{z},y_{\boldsymbol{z}}) \sim R}[\ell(\psi_{\boldsymbol{\theta}}(\boldsymbol{z}), y_{\boldsymbol{z}})] - \ell((\psi_{\boldsymbol{\theta}}(\boldsymbol{z}), y_{\boldsymbol{z}})\right] \\
\leq&\mathbb{E}\left[\sup_{\psi_{\boldsymbol{\theta}} \in \psi} (\mathbb{E}_{(\mathbf{x},\bar{y}) \sim \mathcal{D}}[\ell(\psi_{\boldsymbol{\theta}}(\mathbf{x}), \bar{y})] - \mathbb{E}_{(\boldsymbol{z},y_{\boldsymbol{z}}) \sim R}[\ell(\psi_{\boldsymbol{\theta}}(\boldsymbol{z}), y_{\boldsymbol{z}})]) + \sup_{\psi_{\boldsymbol{\theta}} \in \psi} (\mathbb{E}_{(\boldsymbol{z},y_{\boldsymbol{z}}) \sim R}[\ell(\psi_{\boldsymbol{\theta}}(\boldsymbol{z}), y_{\boldsymbol{z}})] - \ell((\psi_{\boldsymbol{\theta}}(\boldsymbol{z}), y_{\boldsymbol{z}}))\right] \\
\leq&disc_H(\mathcal{D}, R) + \sup_{\psi_{\boldsymbol{\theta}} \in \psi} \mathbb{E}_{(\boldsymbol{z},y_{\boldsymbol{z}}) \sim R}[\ell(\psi_{\boldsymbol{\theta}}(\boldsymbol{z}), y_{\boldsymbol{z}})] - \ell((\psi_{\boldsymbol{\theta}}(\boldsymbol{z}), y_{\boldsymbol{z}})] \\
\leq&disc_H(\mathcal{D}, R) + 2\Re(\psi, \ell_q)
\end{aligned} \tag{17}$$

where $\Re(\psi, \ell_q)$ is the empirical Rademacher complexity of the loss function $\ell_q$ and $\ell_q$ is the $\ell_q$ loss for some $q \geq 1$, $\ell(\psi_{\theta}(\mathbf{x}, \bar{y})) = |\bar{y}' - \bar{y}|^q$ for $\bar{y}, \bar{y}' \in \mathcal{Y}$. It can be expressed as the upper bound of the hypothesis set $\mathcal{H} \subset \{\psi : \mathcal{X} \to \mathcal{Y}\}$. From the (Mohri & Muñoz Medina, 2012), the Rademacher complexity $\Re(\psi, \ell_q) \leq qM^{q-1}\Re(\psi)$, where $\Re(\psi)$ is given as:

$$\Re(\psi) = \mathbb{E}_{\sigma}\left[\sup_{\psi_{\theta} \in \psi} \sigma\ell(\psi_{\theta}(\mathbf{x}), y)\right]. \tag{18}$$

Therefore, by combining Eq.(16), (17) and (18), the upper bound of $\mathcal{L}_{dis}$ can be expressed as:

$$\ell(\psi_{\theta}(\boldsymbol{z}), y_{\boldsymbol{z}}) - \ell(\psi_{\theta}(\mathbf{x}), y) \leq 2qM^{q-1}\Re(\psi) + disc_{\mathcal{H}}(\mathcal{D}, R) + M\sqrt{\frac{\log \frac{1}{\epsilon}}{2}}. \tag{19}$$

## B.2 The upper bound of $\mathcal{L}_{MMD}$

First, we perform the upper bound analysis for $\mathcal{L}_{MMD}$. Based on Definition 1 of (Mohri & Muñoz Medina, 2012), we can obtain the Rademacher complexity of the network embedding function $\psi_{\boldsymbol{\vartheta}}$ with respect to $|\mathcal{D}|$-sample. The specific expression is

$$\Re_{\mathcal{D}}(\psi) = \mathbb{E}_{\boldsymbol{\sigma}}\left[\sup_{\psi_{\boldsymbol{\vartheta}} \in \psi} \frac{1}{|\mathcal{D}|} \sum_{i=1}^{|\mathcal{D}|} \sigma_i \psi_{\boldsymbol{\vartheta}}(\mathbf{x}_i)\right], \tag{20}$$

where $\boldsymbol{\sigma} = (\sigma_i, \ldots, \sigma_{\mathcal{D}})^\top$, with $\sigma_i$ independent uniform random variables taking values in $\{-1, 1\}$. According to (Golowich et al., 2018), let $\psi$ be the embedding network of depth $l$ over the topological $\mathcal{X}$, where each network parameter matrix $\boldsymbol{\vartheta}_j$ has Frobenius norm at most $M_F(j)$, and the activation function be a 1-Lipschitz and positve-homogeneous activation function (such as the $ReLU$). Then,

$$\Re_{\mathcal{D}}(\psi) \leq \frac{B(\sqrt{2\log(2)l} + 1) \prod_{j=1}^{l} M_F(j)}{\sqrt{|\mathcal{D}|}}. \tag{21}$$

where assuming the input $\|\mathbf{x}\| \leq B$. Let $\mathbf{x}$ and $\boldsymbol{z}$ be random variables defined on a topological space $\mathcal{X}$, with respective Borel probability measures $p$ and $q$. Given observations $\mathcal{D} := \{\mathbf{x_1}, \ldots, \mathbf{x}_n\}$ and $R := \{\boldsymbol{z}_1, \ldots, \boldsymbol{z}_m\}$, independently and identically distributed ($i.i.d$) for $p$ and $q$, respectively. Moreover, we use the shorthand notation $\mathbb{E}_{\mathbf{x}}[\psi_{\boldsymbol{\vartheta}}(\mathbf{x})] := \mathbb{E}_{\mathbf{x}\sim p}[\psi_{\boldsymbol{\vartheta}}(\mathbf{x})]$ and $\mathbb{E}_{\boldsymbol{z}}[\psi_{\boldsymbol{\vartheta}}(\boldsymbol{z})] := \mathbb{E}_{\boldsymbol{z}\sim q}[\psi_{\boldsymbol{\vartheta}}(\boldsymbol{z})]$ to denote expectations with respect to $p$ and $q$, respectively, where $\mathbf{x} \sim p$ denotes $\mathbf{x}$ has distribution $p$. The $\psi : \mathbb{R}^d \to \mathbb{R}^{d'}$, where $\mathbb{R}^d$ is the toplogical space $\mathcal{X}$, $d \ll d'$ and $\boldsymbol{\vartheta}$ is the function parameter. Based on the above definition, the Maximum Mean Discrepancy (MMD) can be expressed as:

$$MMD[\psi, p, q] := \sup_{\psi_{\boldsymbol{\vartheta}} \in \psi} (\mathbb{E}_{\mathbf{x}}[\psi_{\boldsymbol{\vartheta}}(\mathbf{x})] - \mathbb{E}_{\boldsymbol{z}}[\psi_{\boldsymbol{\vartheta}}(\boldsymbol{z})]). \tag{22}$$

A biased empirical estimate of the MMD (Gretton et al., 2006) is

$$MMD_b[\psi, \mathcal{D}, R] := \sup_{\psi_{\boldsymbol{\vartheta}} \in \psi} \left( \frac{1}{|\mathcal{D}|} \sum_{i=1}^{|\mathcal{D}|} \psi_{\boldsymbol{\vartheta}}(\mathbf{x}_i) - \frac{1}{|R|} \sum_{j=1}^{|R|} \psi_{\boldsymbol{\vartheta}}(\boldsymbol{z}_j) \right). \tag{23}$$

Furthermore, according to (Garriga-Alonso et al., 2019), the output of a neural network $\psi_{\boldsymbol{\vartheta}}$ (CNN) can be viewed as a Gaussian process under the constraints of an infinite number of (convolutional) filters. Assume the $p$ and $q$ are different, the upper bound on the absolute difference between $MMD[\psi, p, q]$ and $MMD_b[\psi, \mathcal{D}, R]$ is expressed as:

$$
\begin{aligned}
&|MMD[\psi, p, q] - MMD_b[\psi, \mathcal{D}, R]| \\
&= \left| \sup_{\psi_{\boldsymbol{\vartheta}} \in \psi} (\mathbb{E}_p(\psi_{\boldsymbol{\vartheta}}) - \mathbb{E}_q(\psi_{\boldsymbol{\vartheta}})) - \sup_{\psi_{\boldsymbol{\vartheta}} \in \psi} \left( \frac{1}{|\mathcal{D}|} \sum_{i=1}^{|\mathcal{D}|} \psi_{\boldsymbol{\vartheta}}(\mathbf{x}_i) - \frac{1}{|R|} \sum_{j=1}^{|R|} \psi_{\boldsymbol{\vartheta}}(\boldsymbol{z}_j) \right) \right| \\
&\leq \sup_{\psi_{\boldsymbol{\vartheta}} \in \psi} \underbrace{\left| (\mathbb{E}_p(\psi_{\boldsymbol{\vartheta}}) - \mathbb{E}_q(\psi_{\boldsymbol{\vartheta}})) - \frac{1}{|\mathcal{D}|} \sum_{i=1}^{|\mathcal{D}|} \psi_{\boldsymbol{\vartheta}}(\mathbf{x}_i) + \frac{1}{|R|} \sum_{j=1}^{|R|} \psi_{\boldsymbol{\vartheta}}(\boldsymbol{z}_j) \right|}_{\triangle(p, q, \mathcal{D}, R)}
\end{aligned}
\tag{24}
$$

Next, we provide an upper bound on the difference between $\triangle(p, q, \mathcal{D}, R)$ and its expectation. Changing either of $\mathbf{x}$ or $\boldsymbol{z}$ in $\triangle(p, q, \mathcal{D}, R)$ results in changes in magnitude of at most $2\Re_{\mathcal{D}}(\psi)/|\mathcal{D}|^{\frac{1}{2}}$ or $2\Re_R(\psi)/|R|^{\frac{1}{2}}$, respectively. We can apply the McDiarmid's inequality theorem (McDiarmid, 1989) to obtain

$$\Pr\left( \triangle(p, q, \mathcal{D}, R) - 2\mathbb{E}_{\mathcal{D}, R}[\triangle(p, q, \mathcal{D}, R)] > \epsilon \right) \leq \exp\left(-\frac{\epsilon|\mathcal{D}||R|}{2\Re_{\mathcal{D}}(\psi)^2 \Re_R(\psi)^2(|\mathcal{D}| + |R|)}\right), \tag{25}$$

where $\epsilon$ is the every arbitrary small and $\epsilon > 0$. In addition, we exploit symmetrization to upper bound the expectation of $\triangle(p, q, \mathcal{D}, R)$. we have

$$
\begin{aligned}
&\mathbb{E}_{\mathcal{D},R}[\triangle(p, q, \mathcal{D}, R)] \\
=&\mathbb{E}_{\mathcal{D},R} \sup_{\psi_{\boldsymbol{\vartheta}} \in \psi} \left| \mathbb{E}_p(\psi_{\boldsymbol{\vartheta}}) - \frac{1}{|\mathcal{D}|} \sum_{i=1}^{|\mathcal{D}|} \psi_{\boldsymbol{\vartheta}}(\mathbf{x}_i) - \mathbb{E}_q(\psi_{\boldsymbol{\vartheta}}) + \frac{1}{|R|} \sum_{j=1}^{|R|} \psi_{\boldsymbol{\vartheta}}(\mathbf{z}_j) \right| \\
=&\mathbb{E}_{\mathcal{D},R} \sup_{\psi_{\boldsymbol{\vartheta}} \in \psi} \left| \mathbb{E}_{\mathcal{D}'}(\psi_{\boldsymbol{\vartheta}}) - \frac{1}{|\mathcal{D}|} \sum_{i=1}^{|\mathcal{D}|} \psi_{\boldsymbol{\vartheta}}(\mathbf{x}_i) - \mathbb{E}_{R'}(\psi_{\boldsymbol{\vartheta}}) + \frac{1}{|R|} \sum_{j=1}^{|R|} \psi_{\boldsymbol{\vartheta}}(\mathbf{z}_j) \right| \\
\underset{(a)}{\leq}&\mathbb{E}_{\mathcal{D},R,\mathcal{D}',R'} \sup_{\psi_{\boldsymbol{\vartheta}} \in \psi} \left| \frac{1}{|\mathcal{D}|} \sum_{i=1}^{|\mathcal{D}|} \psi_{\boldsymbol{\vartheta}}(\mathbf{x}_i') - \frac{1}{|\mathcal{D}|} \sum_{i=1}^{|\mathcal{D}|} \psi_{\boldsymbol{\vartheta}}(\mathbf{x}_i) - \frac{1}{|R|} \sum_{j=1}^{|R|} \psi_{\boldsymbol{\vartheta}}(\mathbf{z}_j') + \frac{1}{|R|} \sum_{j=1}^{|R|} \psi_{\boldsymbol{\vartheta}}(\mathbf{z}_j) \right| \\
=&\mathbb{E}_{\mathcal{D},R,\mathcal{D}',R',\sigma,\sigma'} \sup_{\psi_{\boldsymbol{\vartheta}} \in \psi} \left| \frac{1}{|\mathcal{D}|} \sum_{i=1}^{|\mathcal{D}|} \sigma_i(\psi_{\boldsymbol{\vartheta}}(\mathbf{x}_i') - \psi_{\boldsymbol{\vartheta}}(\mathbf{x}_i)) + \frac{1}{|R|} \sum_{j=1}^{|R|} \sigma_j'(\psi_{\boldsymbol{\vartheta}}(\mathbf{z}_j') - \psi_{\boldsymbol{\vartheta}}(\mathbf{z}_j)) \right| \\
\underset{(b)}{\leq}&\mathbb{E}_{\mathcal{D},\mathcal{D}',\sigma} \sup_{\psi_{\boldsymbol{\vartheta}} \in \psi} \left| \frac{1}{|\mathcal{D}|} \sum_{i=1}^{|\mathcal{D}|} \sigma_i(\psi_{\boldsymbol{\vartheta}}(\mathbf{x}_i') - \psi_{\boldsymbol{\vartheta}}(\mathbf{x}_i)) \right| + \mathbb{E}_{R,R',\sigma'} \sup_{\psi_{\boldsymbol{\vartheta}} \in \psi} \left| \frac{1}{|R|} \sum_{j=1}^{|R|} \sigma_j'(\psi_{\boldsymbol{\vartheta}}(\mathbf{z}_j') - \psi_{\boldsymbol{\vartheta}}(\mathbf{z}_j)) \right| \\
\underset{(c)}{\leq}&2[\Re_{\mathcal{D}}(\psi, p) + \Re_R(\psi, q)].
\end{aligned}
\tag{26}
$$

where $(a)$ uses Jensen's inequality, $(b)$ uses the triangle inequality, $(c)$ substitutes Definition 20. By combining Eq.(24) and Eq.(26), we derive the final upper bound result, specifically given by:

$$
\Pr\left(\triangle(p, q, \mathcal{D}, R) - 2[\Re_{\mathcal{D}}(\psi, p) + \Re_R(\psi, q)] > \epsilon\right) \leq \exp(-\frac{\epsilon|\mathcal{D}||R|}{2\Re_{\mathcal{D}}(\psi)^2 \Re_R(\psi)^2 (|\mathcal{D}| + |R|)})
\tag{27}
$$

Therefore, the bound of $\mathcal{L}_{MMD}$ is expressed as:

$$
\mathcal{L}_{MMD}(\mathcal{D}, R) \leq 2\Re_{\mathcal{D}} + 2\Re_R + \Re_{\mathcal{D}}\Re_R \sqrt{\frac{(|\mathcal{D}| + |R|) \log \frac{1}{\epsilon}}{2|\mathcal{D}||R|}}
\tag{28}
$$

where the definition of $\Re(\psi, p)$ is similar to $\Re(\psi)$ by giving in Eq.(21).

## C  Proof of Proposition 4.5.

Based on the Assumption 4.1, we assume that the random neural network $\psi_{\boldsymbol{\theta}} : \mathbb{R}^d \to \mathbb{R}^1$ to be the 2-layer network with Binary Cross-Entropy Loss and the random neural embedding network $\psi_{\boldsymbol{\vartheta}} : \mathbb{R}^d \to \mathbb{R}^{d'}$ to be the 2-layer network for the sake of inference and analysis. The activation function in both networks is set to *ReLU*, denoted as $\rho$. Thus, from a specific class, the objective function of the DALI approach, represented by Eq.(4), can be rewritten as:

$$
g(\underbrace{\frac{1}{|\mathcal{D}|} \sum_{i=1}^{|\mathcal{D}|} \ell_{0-1}(\rho(\boldsymbol{\theta}, \mathbf{x}_i), \bar{y}_i)) - \frac{1}{|R|} \sum_{j=1}^{|R|} \ell_{0-1}(\rho(\boldsymbol{\theta}, \mathbf{z}_j), y_{\mathbf{z}_j}))}_{\mathbf{d}_{dis}} + \underbrace{\frac{1}{|\mathcal{D}|} \sum_{i=1}^{|\mathcal{D}|} \rho(\boldsymbol{\vartheta} \cdot \mathbf{x}_i) - \frac{1}{|R|} \sum_{j=1}^{|R|} \rho(\boldsymbol{\vartheta} \cdot \mathbf{z}_i)}_{\mathbf{d}_{MMD}},
\tag{29}
$$

where

$$
\ell_{0-1}(\mathbf{x}_i, \bar{y}_i) = \bar{y}_i \log \Theta(\mathbf{x}_i) + (1 - \bar{y}_i) \log(1 - \Theta(\mathbf{x}_i))
\tag{30}
$$

and $\Theta(\mathbf{x}) = 1/(1 + \exp(-\mathbf{x}))$ is the sigmoid activate function for binary classification.

Since each element $\theta_{i,j}$ of the $i$-th layer network parameter $\boldsymbol{\theta}_i$ exists as $\theta_{i,j} \overset{i.i.d}{\sim} \mathcal{N}(0, 1)$, for a given input vector $\mathbf{x} = [x_j]_{1 \leq j \leq d} \in \mathbb{R}^d$, the operation of this network layer without activation function is given as (Dong et al., 2022):

$$
\mathbf{h} = \boldsymbol{\theta} \cdot \mathbf{x} = [\sum_{j=1}^d \theta_{i,j} x_j]_{i \leq i \leq d'} = [h_i]_{i \leq i \leq d'} \in \mathbb{R}^k,
\tag{31}
$$

where $h_i \overset{i.i.d}{\sim} \mathcal{N}(0, \sum_{j=1}^d x_j^2)$. Then, since activation function $\rho(x) := \max(0, x)$, $\rho(\mathbf{h}) = [\max(0, h_i)]_{i \le i \le k} \in \mathbb{R}^k$. Define $Y = \max(0, X)$, where the random variable $X \sim \mathcal{N}(0, \sigma^2)$. Moreover, $H$ is following the same distribution of $B|X|$, where $B \sim Bernoulli(\frac{1}{2})$ indenpent of $X$ and $\mathbb{E}_X[H] = \mathbb{E}_B[B]\mathbb{E}_X[|X|]$. As a result, for each input of activation function, we have the corresponding ouput is given as

$$\rho(\mathbf{y}) = \rho(\boldsymbol{\theta}\mathbf{x}) = \mathbf{B} \odot |\boldsymbol{\theta}\mathbf{x}|, \tag{32}$$

where $\odot$ is element-wise multiplication and $\mathbf{B} = [B_i]_{1 \le i \le k}$. By substituting the above formula into Eq.(29), we can obtain further rewriting of $\mathbf{d}_{dis}$ and $\mathbf{d}_{MMD}$, which are expressed as:

$$\mathbf{d}_{dis} = \frac{1}{|\mathcal{D}|}\sum_{i=1}^{|\mathcal{D}|} \ell_{0-1}(\boldsymbol{\theta}B_i^x \mathrm{sgn}(\boldsymbol{\theta}\mathbf{x}_i)\mathbf{x}_i, \bar{y}_i) - \frac{1}{|R|}\sum_{j=1}^{|R|} \ell_{0-1}(\boldsymbol{\theta}B_j^z \mathrm{sgn}(\boldsymbol{\theta}\mathbf{z}_j)\mathbf{z}_j, y_{\mathbf{z}_j}) \tag{33}$$

and

$$\mathbf{d}_{MMD} = \boldsymbol{\vartheta}\left(\frac{1}{|\mathcal{D}|}\sum_{i=1}^{|\mathcal{D}|} B_i^x \mathrm{sgn}(\boldsymbol{\vartheta}\mathbf{x}_i)\mathbf{x}_i - \frac{1}{|R|}\sum_{j=1}^{|R|} B_j^z \mathrm{sgn}(\boldsymbol{\vartheta}\mathbf{z}_j)\mathbf{z}_j\right). \tag{34}$$

In the above of Eq.(36) and Eq.(37), $\mathrm{sgn}(\mathbf{x})$ denote the sign of $\mathbf{x}$ that $|\mathbf{x}| = \mathrm{sgn}(\mathbf{x})\mathbf{x}$. Since this focuses on a specific class, the vector of Bernoulli random variable $\mathbf{B}_i^x$ reduces to the single random variable $B_i^x$.

To simplify the derivation, an auxiliary variable $\varphi$ is introduced, defined as:

$$\varphi_{\boldsymbol{\theta},\mathbf{x}_i} = B_i^x \mathrm{sgn}(\boldsymbol{\theta}\mathbf{x}_i). \tag{35}$$

The $\mathbf{d}_{dis}$ and $\mathbf{d}_{MMD}$ can be rewritten as:

$$\begin{aligned}\mathbf{d}_{dis} =& \frac{1}{|\mathcal{D}|}\sum_{i=1}^{|\mathcal{D}|} \ell_{0-1}(\varphi_{\boldsymbol{\theta},\mathbf{x}_i}\boldsymbol{\theta}\mathbf{x}_i)\mathbf{x}_i, \bar{y}_i) - \frac{1}{|R|}\sum_{j=1}^{|R|} \ell_{0-1}(\varphi_{\boldsymbol{\theta},\mathbf{z}_j}\boldsymbol{\theta}\mathbf{z}_j, y_{\mathbf{z}_j}) \\ =& \frac{1}{|\mathcal{D}|}\sum_{i=1}^{|\mathcal{D}|}(\bar{y}_i\log\Theta(\varphi_{\boldsymbol{\theta},\mathbf{x}_i}\boldsymbol{\theta}\mathbf{x}_i) + (1-y_i)\log(1-\Theta(\varphi_{\boldsymbol{\theta},\mathbf{x}_i}\boldsymbol{\theta}\mathbf{x}_i))) \\ & -\frac{1}{|R|}\sum_{j=1}^{|R|}(y_{\mathbf{z}_j}\log\Theta(\varphi_{\boldsymbol{\theta},\mathbf{z}_j}\boldsymbol{\theta}\mathbf{z}_j) + (1-y_{\mathbf{z}_j})\log(1-\Theta(\varphi_{\boldsymbol{\theta},\mathbf{z}_j}\boldsymbol{\theta}\mathbf{z}_j)))\end{aligned} \tag{36}$$

and

$$\mathbf{d}_{MMD} = \boldsymbol{\vartheta}\left(\frac{1}{|\mathcal{D}|}\sum_{i=1}^{|\mathcal{D}|} \varphi_{\boldsymbol{\vartheta},\mathbf{x}_i}\mathbf{x}_i - \frac{1}{|R|}\sum_{j=1}^{|R|} \varphi_{\boldsymbol{\vartheta},\mathbf{z}_j}\mathbf{z}_j\right). \tag{37}$$

For each specification data point $\mathbf{z}_s$, the derivatives $\mathcal{L}_{dis}$ and $\mathcal{L}_{MMD}$ are given as:

$$\frac{\partial\mathcal{L}_{dis}}{\partial\mathbf{z}_s} = \mathbb{E}_{\boldsymbol{\theta}}\left[\frac{\mathbf{d}_{dis}}{\sqrt{\mathbf{d}_{dis}^2 + \alpha}}\frac{\partial\mathbf{d}_{dis}}{\partial\mathbf{z}_s}\right] \tag{38}$$

and

$$\frac{\partial\mathcal{L}_{MMD}}{\partial\mathbf{z}_s} = \mathbb{E}_{\boldsymbol{\vartheta}}\left[\frac{(\mathbf{d}_{MMD})^2}{\partial\mathbf{z}_s}\right] = \mathbb{E}_{\boldsymbol{\vartheta}}\left[2(\frac{\partial\mathbf{d}_{MMD}}{\partial\mathbf{z}_s})\mathbf{d}_{MMD}\right], \tag{39}$$

where

$$\frac{\partial\mathbf{d}_{dis}}{\partial\mathbf{z}_s} = -\frac{1}{|R|}(y_{\mathbf{z}_s} - \Theta(\varphi_{\boldsymbol{\theta},\mathbf{z}_s}\boldsymbol{\theta}\mathbf{z}_s))(\varphi_{\boldsymbol{\theta},\mathbf{z}_s}\boldsymbol{\theta}^\top) \tag{40}$$

and

$$\frac{\partial\mathbf{d}_{MMD}}{\partial\mathbf{z}_s} = -\frac{1}{|R|}\varphi_{\boldsymbol{\vartheta},\mathbf{x}_i}\boldsymbol{\vartheta}^\top. \tag{41}$$

Therefore, the gradient of $\mathcal{L}_{dis}$ and $\mathcal{L}_{MMD}$ on $\boldsymbol{z_s}$ are given as:

$$\frac{\partial \mathcal{L}_{dis}}{\partial \boldsymbol{z}_s} = \mathbb{E}_{\boldsymbol{\theta}}\left[\frac{\mathbf{d}_{dis}}{\sqrt{\mathbf{d}_{dis}^2 + \alpha}}(-\frac{1}{|R|}(y_{\boldsymbol{z}_s} - \Theta(\varphi_{\boldsymbol{\theta},\boldsymbol{z}_s}\boldsymbol{\theta}\boldsymbol{z}_s))(\varphi_{\boldsymbol{\theta},\boldsymbol{z}_s}\boldsymbol{\theta}^\top))\right]$$

$$= \mathbb{E}_{\boldsymbol{\theta}}\left[\frac{\mathbf{d}_{dis}}{\sqrt{\mathbf{d}_{dis}^2 + \alpha}}(-\frac{1}{|R|}(y_{\boldsymbol{z}_s} - \Theta(B_s^z\mathrm{sgn}(\boldsymbol{\theta}\boldsymbol{z}_s)\boldsymbol{\theta}\boldsymbol{z}_s))(B_s^z\mathrm{sgn}(\boldsymbol{\theta}\boldsymbol{z}_s)\boldsymbol{\theta}^\top))\right], \quad (42)$$

and

$$\frac{\partial \mathcal{L}_{MMD}}{\partial \boldsymbol{z}_s}$$

$$= \mathbb{E}_{\boldsymbol{\vartheta}}\left[2(-\frac{1}{|R|}\varphi_{\boldsymbol{\vartheta},\boldsymbol{z}_s}\boldsymbol{\vartheta}^\top)\mathbf{d}_{MMD}\right]$$

$$= \mathbb{E}_{\boldsymbol{\vartheta}}\left[-\frac{2}{|R|}\boldsymbol{\vartheta}^\top\boldsymbol{\vartheta}(\frac{1}{|\mathcal{D}|}\sum_{i=1}^{|\mathcal{D}|}\varphi_{\boldsymbol{\vartheta},\boldsymbol{z}_s}\varphi_{\boldsymbol{\vartheta},\mathbf{x}_i}\mathbf{x}_i - \frac{1}{|R|}\sum_{j=1}^{|R|}\varphi_{\boldsymbol{\vartheta},\boldsymbol{z}_s}\varphi_{\boldsymbol{\vartheta},\boldsymbol{z}_j}\boldsymbol{z}_j)\right] \quad (43)$$

$$= \mathbb{E}_{\boldsymbol{\vartheta}}\left[-\frac{2}{|R|}\boldsymbol{\vartheta}^\top\boldsymbol{\vartheta}(\frac{1}{|\mathcal{D}|}\sum_{i=1}^{|\mathcal{D}|}B_s^z\mathrm{sgn}(\boldsymbol{\vartheta}\boldsymbol{z}_s)B_i^x\mathrm{sgn}(\boldsymbol{\vartheta}\mathbf{x}_i)\mathbf{x}_i - \frac{1}{|R|}\sum_{j=1}^{|R|}B_s^z\mathrm{sgn}(\boldsymbol{\vartheta}\boldsymbol{z}_s)B_j^z\mathrm{sgn}(\boldsymbol{\vartheta}\boldsymbol{z}_j)\boldsymbol{z}_j)\right].$$

According to (Dong et al., 2022), we know that $\mathbb{E}_{\boldsymbol{\vartheta}}[\mathrm{sgn}(\boldsymbol{\vartheta}\mathbf{x})\mathrm{sgn}(\boldsymbol{\vartheta}\boldsymbol{z})\boldsymbol{\vartheta}^\top\boldsymbol{\vartheta}] \in \mathbb{R}^{d\times d}$ and

$$\mathbf{x}^\top\mathbb{E}_{\boldsymbol{\vartheta}}[\mathrm{sgn}(\boldsymbol{\vartheta}\mathbf{x})\mathrm{sgn}(\boldsymbol{\vartheta}\boldsymbol{z})\boldsymbol{\vartheta}^\top\boldsymbol{\vartheta}]\boldsymbol{z} = \mathbb{E}_{\boldsymbol{\vartheta}}[(\mathrm{sgn}(\boldsymbol{\vartheta}\mathbf{x})\boldsymbol{\vartheta}\mathbf{x})^\top\mathrm{sgn}(\boldsymbol{\vartheta}\boldsymbol{z}_j)\boldsymbol{\vartheta}\boldsymbol{z}_j] = \frac{\|\mathbf{x}\|_2\|\boldsymbol{z}\|_2}{\pi}[(\pi - 2\phi)cos(\phi) + 2sin(\phi)], \quad (44)$$

where if $\mathbf{x} = \boldsymbol{z}$, $\mathbb{E}_{\boldsymbol{\vartheta}}[\mathrm{sgn}(\boldsymbol{\vartheta}\mathbf{x})\mathrm{sgn}(\boldsymbol{\vartheta}\boldsymbol{z})\boldsymbol{\vartheta}^\top\boldsymbol{\vartheta}] = \mathbf{I}_d$, while $\mathbb{E}_{\boldsymbol{\vartheta}}[\mathrm{sgn}(\boldsymbol{\vartheta}\mathbf{x})\mathrm{sgn}(\boldsymbol{\vartheta}\boldsymbol{z})\boldsymbol{\vartheta}^\top\boldsymbol{\vartheta}] = -\mathbf{I}_d$ if $\mathbf{x} = -\boldsymbol{z}$. In fact, this formula can be seen as a matrix depending on the angle $\phi$ between $\mathbf{x}$ and $\boldsymbol{z}$.

In summary, in the DALI appoach, the initialization of $\boldsymbol{z}$ is randomly selected from $\mathcal{D}$, ensuring that $\mathbf{x} \sim \boldsymbol{z}$, $\mathcal{D} \sim R$ and sharing corresponding labels. Based on Eq.(42) and Eq.(43), it can be concluded that even though $\boldsymbol{z}$ and $\mathbf{x}$ undergo transformations through two layers of a non-linear neural network, their classification performance and mean values can still be preserved. During optimization, the aforementioned equations also indicate that each $\boldsymbol{z}$ is updated in the direction that minimizes the task performance discrepancy and the pseudo-barycenter distance between datasets. Therefore, the specification $R$ not only aligns the class feature distributions but also aligns the discriminative ability during the optimization process, ensuring that the specification can more comprehensively skecth the model's capability.

The above reasoning is based on the assumption of a two-layer non-linear network. Extending this analysis to a multi-layer

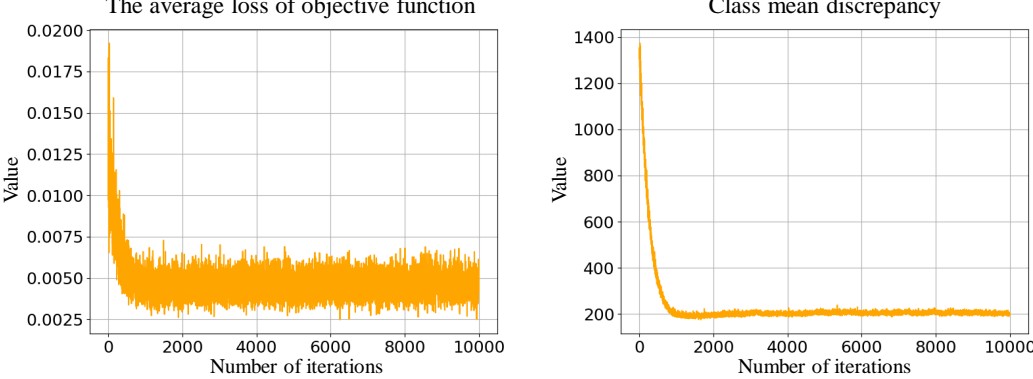

Figure 3. During the optimization iterations, the average loss value of the objective function for the DALI approach (left) and the class mean discrepancy between the optimized specification and the training data (right).

non-linear network still holds. If $\forall z \in \{z_i | (z_i, y_{z_i}) \in R\}$, $\exists \mathbf{x} \in \{(\mathbf{x}_i, y_i) \in \mathcal{D}\}$, $z = \mathbf{x}$ and $y, y_z \in \mathcal{H}$, we have

$$
\begin{aligned}
&\mathcal{L}(\boldsymbol{\theta}_{l+1}(\rho_l \cdots (\rho_1(\boldsymbol{\theta}_1 \cdot \mathbf{x}))), \bar{y}) - \mathcal{L}(\boldsymbol{\theta}_{l+1}(\rho_l \cdots (\rho_1(\boldsymbol{\theta}_1 \cdot \boldsymbol{z}^*))), y_{\boldsymbol{z}}) + \\
&\frac{1}{|\mathcal{D}|} \sum_{n=1}^{|\mathcal{D}|} \rho_l(\boldsymbol{\vartheta}_l \cdots (\rho_1(\boldsymbol{\vartheta}_1 \cdot \mathbf{x}_n))) - \frac{1}{|R|} \sum_{m=1}^{|R|} \rho_l(\boldsymbol{\vartheta}_l \cdots (\rho_1(\boldsymbol{\vartheta}_1 \cdot \boldsymbol{z}_m^*))) \to 0.
\end{aligned}
\tag{45}
$$

Additionally, we provide auxiliary proof through experiments. In this case, data from label space A in the autumn domain of the NICO dataset is selected as the task training data to generate specification. The training dataset comprises 10 classes, and the specification size is set to $10 \times 5$. Throughout this process, we continuously record the average loss value of the objective function for the DALI approach at each optimization iteration and the class mean discrepancy between the optimized specification and the training dataset. Details are illustrated in the Figure 3. The experimental results show that the DALI approach's objective function reaches optimal performance after approximately 1000 iterations, while the class mean discrepancy between the specification and the training data remains in the most similar state. This further confirms the class pseudo-barycenter relationship between the specification and the training dataset, while also supporting the privacy protection analysis of the DALI approach.

## D  Proof of Proposition 4.6

Suppose the number of training dataset $D$ classes is one. Since the size of the specification $R$, denoted as $Km$, is significantly smaller than the size of training dataset $\mathcal{D}$, denoted as $n$, and based on Assumption 4.2 and Proposition 4.5, in the objective function's optimization process of the DALI specification approach, we can obtain:

$$
\boldsymbol{z}_i^* = \mathbf{x}_j + \ell(\psi_{\boldsymbol{\theta}}(\mathbf{x}), \bar{y}) - \ell(\psi_{\boldsymbol{\theta}}(\boldsymbol{z}), y_{\boldsymbol{z}}) + \frac{1}{|\mathcal{D}|} \sum_{i=1}^{|\mathcal{D}|} \psi_{\boldsymbol{\vartheta}}(\mathbf{x}_i) - \frac{1}{|R|} \sum_{j=1}^{|R|} \psi_{\boldsymbol{\vartheta}}(\boldsymbol{z}_j)
\tag{46}
$$

where the specification $R$ is initialized with first $|R|$ samples of $\mathcal{D}$, i.e, $\forall z \in \{z_i | (z_i, y_{z_i}) \in R\}$, $\exists \mathbf{x} \in \{(\mathbf{x}_i, y_i) \in \mathcal{D}\}$, $z = \mathbf{x}$. Evidently, in the DALI approach, the discrepancy between the initial specification and the pseudo-barycenter of the training data decreases as the specification size increases. Conversely, the discrepancy becomes more pronounced when the specification size is smaller. This offers a compelling rationale for why a larger specification size could lead to data leakage. In practice, due to differences in data size, the mean and classification performance of randomly sampled specification and the training data are different. Consequently, the optimized $\boldsymbol{z}_i^*$ still exhibits certain differences in discriminative alignment and distribution alignment. This ensures the ability of privacy protection.

## E  Implementation details

To validate that the proposed DALI approach can generate high-quality specification in the learnware paradigm, we use RKME, RKME-W, and LANE specification methods as baselines. Detailed descriptions of these methods are provided as follows:

- RKME (Wu et al., 2023): Using Kernel Mean Embedding (KME), a reduced set is generated in the Reproducing Kernel Hilbert Space (RKHS) via Maximum Mean Discrepancy (MMD) as the specification.

- RKME-W (Guo et al., 2023): Building upon RKME, class-specific reduced model parameters are incorporated into the reduced set to form the specification.

- LANE (Chen et al., 2025): In the random neural embedding space, specification is generated by aligning with the inter-class feature distribution of the training data.

In the comparative experiments, the kernel functions of RKME and RKME-W are Gaussian kernels with a bandwidth parameter of 2.0, and the reduced model of RKME-W achieves optimal parameters through *ResNet-18* (He et al., 2016). Furthermore, the parameters involved in the LANE method and the proposed DALI approach are identical, including a batch size set to 64, a *ConvNetBN* (Rawat & Wang, 2017) network architecture, an activation function set to *ReLU*, and a normalization layer set to *GroupNorm*.

