# OpenReview forum: "Learnware Specification via Dual Alignment"
_ICML.cc/2025/Conference — ICML 2025 poster_

### Official Review · Reviewer_2dDo · 2025-03-12

**Overall Recommendation:** 4

**Summary:**

The learnware system is a model reuse system which is designed to choose the optimal model from a model repository based on rules derived from user datasets. The core of this system lies in the use of specifications for model selection. This paper introduces a novel specifications generation method called Dual Alignment, which consists of two components: discriminative alignment and distribution alignment. Compared to the traditional RKME-based specifications generation, Dual Alignment demonstrates performance improvements across various metrics.

**Claims And Evidence:**

Yes

**Essential References Not Discussed:**

No

**Experimental Designs Or Analyses:**

In Table 1, the selection of size introduces a hyperparameter K, but its exact definition is unclear. What does K specifically refer to? Do K×5 and K×10 correspond to 20 and 100 in the RKME algorithm?

In Table 2, two metrics are introduced: superclass accuracy and quality. However, their explanations are somewhat abstract, especially superclass accuracy. How is superclass accuracy actually calculated? Also, why can different quality values exist when superclass accuracy is the 100%? This part is a bit confusing.

Additionally, compared to the RKME algorithm, the Dual Alignment method generates labels（check mark on label selection）. Do these labels refer to the pseudo-labels generated during the Submitting Stage?

**Methods And Evaluation Criteria:**

There is an issue with the selection of the evaluation dataset. In Section 5.1, it is mentioned that “we extract 4 label spaces from the overlapping classes of 11 domains across the two datasets.” However, in the later testing in Section 5.3, only label spaces A and B were used. Why were only these two chosen? And how were they selected?

**Other Comments Or Suggestions:**

No

**Other Strengths And Weaknesses:**

Pros:

This paper provides very detailed proofs. It gives upper bounds for both of the proposed loss functions and also includes an analysis of privacy issues in this scenario. From a theoretical perspective, it is a solid and comprehensive paper.

Cons:

The 5.1 Experimental Setup section does not clearly explain the metrics, which might make it difficult for readers who are not familiar with these metrics to fully understand the experiment results.

**Questions For Authors:**

No question

**Relation To Broader Scientific Literature:**

Yes, this paper makes a significant contribution to the learnware paradigm. Previous learnware systems did not distinguish between discriminative ability and feature distribution, but this paper introduces and analyzes them in detail. By bringing in a new perspective, it enhances the overall capability of the learnware system and could serve as valuable inspiration for future research.

The paper introduces a new classification dimension, helping users find the models they need more effectively. Additionally, the performance improvements from multiple analysis dimensions might complement each other which further boosting the system’s effectiveness.I’m curious if there will be any future work on combining this approach with the traditional RKME algorithm.

**Theoretical Claims:**

Yes

---

> ### Author Rebuttal · Authors · 2025-03-31
>
> Thanks for the valuable feedback and appreciation of our work. We hope that our responses could mitigate your concerns.
>
> Q1: Question on the mixed task setting
>
> Ans: In the experiments of this paper, we set up four label spaces. It is no difficult to find that the label space A(B) contains the label space C(D). In the mixed task setting, if the domains corresponding to all label spaces are set as developer tasks, this will lead to duplicate tasks in the dock system. The learngware paradigm for duplicate tasks will only keep the tasks with the best model performance, so here we do not use label space C and D in the mixed tasks setting. These two label spaces are still constructed mainly for heterogeneous label space setting.
>
> Q2: Question on the parameter K.
>
> Ans: The parameter K is the number of dataset classes. If the number of the task dataset classes is 5, the specification size $K\times 5 = 25$. In the latest version, we will further clarify this.
>
> Q3: Questions on the evaluation metrics.
>
> Ans: In the Mixed Task Setting, both the superclass accuracy and quality metrics are actually classification accuracy metrics, just assessed differently.
>
> In this setting, we have 11 domains $\times$ 2 label spaces (A and B) $=$ 22 developer tasks as superclasses, i.e., 22 superclasses; and label space A$\cup$B$\times$11 domains $=$ 11 user requirement tasks. Thus, each requirement task contains two superclasses, i.e., each requirement task requires two corresponding learnware to be solved. This ensures that no single learnware model in the system can solve the requirement independently, but rather, a combination of models is required to address the requirement.
> Furthermore, in order to evaluate the performance of the learnware dock system constructed by each specification method, it is implemented here by evaluating the accuracy of the system in identifying useful learnware, i.e., the superclass accuracy. Meanwhile, our proposed method is to generate a class specification. In order to verify the inter-class discriminability of the class specification, it is evaluated here by accessing the class accuracy of the corresponding specifications through the developer's pre-trained model test, i.e., quality metric. In the latest version, we will further emphasize the description of the metrics.
>
> In addition, since smaller specification size lose more information about the original data, this can lead to poorer  discriminability of the statutes, and thus poorer quality. However, since the deploying stage of the learnware paradigm based on neural embedding specification is identified useful learnware based on mean unbiased estimation, the discriminability of the specification do not greatly  affect the performance of the learnware paradigm (superclass accuracy metrics). This is why the experiments in this paper show the values of different quality metrics with 100% superclass accuracy metrics.
>
> Q4: Question on the label (check mark on label selection) in the Table 2.
>
> Ans:  In Table 2, label (check mark on label selection) refers to the label of the specification, i.e., the real label.The specification generated by the \textsc{Lane} method and the \textsc{Dali} method are class specification with label. According to the previous answer, we know that the assessment of quality metric requires label to be obtained, so the RKME and RKME-W methods do not have label to obtain the quality assessment metric values, whereas the \textsc{Lane} and the \textsc{Dali} can obtain them.

---

### Official Review · Reviewer_jxaa · 2025-03-12

**Overall Recommendation:** 4

**Summary:**

This paper introduces a approach (DALI) to generating high-quality model specifications in the learnware paradigm. Unlike existing methods that rely solely on distribution alignment, DALI incorporates discriminative alignment, which captures the model's intrinsic discriminative performance. By jointly considering both alignments, DALI enhances specification quality, enabling more effective model reuse in a learnware dock system. Theoretical and empirical results demonstrate DALI's superiority in characterizing model capabilities and handling diverse label space scenarios.

**Claims And Evidence:**

Yes.

**Essential References Not Discussed:**

No.

**Experimental Designs Or Analyses:**

Yes, I have checked the experimental designs and analyses.

**Methods And Evaluation Criteria:**

Yes.

**Other Comments Or Suggestions:**

1. To verify the effectiveness of the proposed learnware specification, it is better to provide more experiments on the tabular tasks.
2. More discussion of the learnware specification design can be provided, please see the questions for details.

**Other Strengths And Weaknesses:**

1. The learnware paradigm is useful when handling well-trained models, it is an interesting paradigm and the specification design is a quite important problem.
2. This paper gives the theoretical analysis of the proposed learnware specification.

**Questions For Authors:**

1. Previous paper [Tan et al., 2024a] also encodes the marginal distribution of data and the conditional distribution of model, what is the difference between DALI and the previous work? In the [Zhou and Tan, 2024], specification is generated by the sketching the distribution of feature concatenated with the model output, what is the potential advantage of additionally considering the true labels of the task data, it there any experiments to verify the effectiveness of this design?
2. When the developer's task is easy, the model outputs are more likely to be accurate, and the two objectives are more likely to align. However, when the task is challenging, the objectives can be conflict. Would it be beneficial to separately model the distribution of true labels and model outputs? In designing learnware recommendation rules, both the information of p(Y|X) from true labels and model outputs can be utilized with techniques like class-wise MMD.
3. How does the newly proposed learnware specification perform in tabular tasks?
4. In the discriminative alignment, how about using the embedding network and MMD distance like in the distribution alignment? It is there any experiment to verify the effectiveness of random feature mapping?

**Relation To Broader Scientific Literature:**

Simultaneously consider the true label distribution and the model output distribution to generate the learnware specification maybe helpful, but this should be more carefully discussed and verified by more experiments in different scenarios (like tabular tasks).

**Theoretical Claims:**

Yes, I have checked the theoretical claims.

---

> ### Author Rebuttal · Authors · 2025-03-31
>
> Thanks for the insightful feedback and the interest in our work! We hope our responses can address your concerns.
>
> Q1: Questions on differences with similar works.
>
> Ans: The method proposed by [Tan et al. 2024a] adds conditional distributions from the pre-trained model's output labels to the marginal distribution of the data, enhancing the discriminativeness of the generated specification distributions. This idea is similar to the \textsc{Dali} method, but their differences still exist in the following two aspects:
>
> (1) [Tan et al. 2024a] is an RKME-based method, and the quality of the generated specification heavily depends on the chosen kernel function. In contrast, our proposed \textsc{Dali} method replaces the kernel function with neural embedding, making it more adaptable to complex scenarios.
>
> (2) \textsc{Dali} collaboratively utilizes pseudo labels and true labels to characterize the model’s properties, whereas [Tan et al. 2024a] only leverages pseudo labels to enhance the discriminability of the class specification.
>
> As for the potential advantage of considering the true labels of the task data, existing work [Chen et al. 2025] has demonstrated the effectiveness of using true labels to characterize feature distributions. This paper can be seen as building on that work by additionally considering model discriminative performance through pseudo-labeling, thereby achieving superior specification generation.
>
> Q2: Concerns about conflict of objectives.
>
> Ans: In the learnware paradigm, the dock system stipulates that the performance of all pre-trained models submitted by developers will not be very poor. Therefore, the \textsc{Dali} method we proposed does not face the issue of objective conflict in most scenarios. On the contrary, simultaneously considering pseudo labels and real labels allows the generated specification to more fully characterize the model, thus promoting model search and reuse. Our experimental results clearly verify this. However, if we consider scenarios where the performance of submitted pre-trained models is not restricted, which goes beyond the scope of this paper, personally, I believe that your idea of separately modeling real labels and model outputs is a promising concept for future research, but it needs to be empirically verified in more open environments.
>
> Q3: Questions about the tabular tasks.
>
> Ans: This work is specifically designed for image-based scenarios and is not readily applicable to tabular data. Tabular data exhibits significant differences from image data in terms of feature sparsity, distribution skewness, and feature engineering nuances [Ye et al. 2024]. Handling tabular data may require additional customized modeling designs, which go beyond the scope of this paper. However, exploring the potential of the \textsc{Dali} method for more diverse modalities is an interesting direction for future research.
>
> Q4: Questions about the modeling of discriminative alignment and random feature mapping.
>
> Ans: In the discriminative alignment, inspired by [Konstantinov and Lampert 2019], [Mohri and Munoz 2012], and [Dong et al. 2022], we adopt the $\mathcal{H}$-discrepancy method to measure distribution differences under limited data. The results of the ablation experiments (Table 4) clearly validate the effectiveness of this modeling approach. Furthermore, [Zhao and Bilen. 2022] well demonstrated that random feature mapping can be used as an interpretation of the input data, preserving the data information in a low-dimensional embedding space. Meanwhile,  Appendix C Proof of Proposition 4.5 provides a theoretical demonstration of why discriminant alignment can be as effective as distribution alignment.
>
> [Chen et al. 2025] Chen, W., Mao, J.-X., and Zhang, M.-L. Learnware specification via label-aware neural embedding. In Proceedings of the 39th AAAI Conference on Artificial Intelligence, Philadelphia, Pennsylvania, 2025.
>
> [Ye et al. 2024] Ye, H. J., Liu, S. Y., Cai, H. R., Zhou, Q. L., Zhan, D. C. A closer look at deep learning on tabular data. arXiv preprint arXiv:2407.00956, 2024.
>
> [Konstantinov and Lampert 2019] Konstantinov, N. and Lampert, C. Robust learning from untrusted sources. In Proceedings of the 36th International Conference on Machine Learning, volume 97, pp.3488–3498, 2019.
>
> [Mohri and Munoz 2012] Mohri, M. and Munoz Medina, A. New analysis and algorithm for learning with drifting distributions. In Algorithmic Learning Theory: 23rd International Conference ALT, volume 7568 of Lecture Notes in Computer Science, pp. 124–138,  2012.
>
> [Dong et al. 2022] Dong, T., Zhao, B., and Lyu, L. Privacy for free: How does dataset condensation help privacy? In \textit{Proceedings of the 33rd International Conference on Machine Learning}, volume 162, pp. 5378–5396, 2022.
>
> [Zhao and Bilen. 2022] Zhao, B. and Bilen, H. Dataset condensation with distribution matching. In Proceedings of the IEEE/CVF Winter Conference on Applications of Computer Vision, pp. 6514-6523, 2023.

---

> > ### Comment · Reviewer_jxaa · 2025-04-05
> >
> > Thank you for the author’s detailed response. I would like to revise my score upward.

---

> > > ### Author Response · Authors · 2025-04-06
> > >
> > > Dear Reviewer jxaa：
> > >
> > > Thank you so much for your kind reply and for adjusting the score! We will revise our paper according to the constructive reviews.
> > >
> > > Best
> > >
> > > Authors

---

### Official Review · Reviewer_PFro · 2025-03-13

**Overall Recommendation:** 3

**Summary:**

The paper shows that existing specification methods primarily rely on distribution alignment to generate specifications and introduces DALI, which incorporates both discriminative and distribution alignments in the process. Theoretical and empirical results demonstrate that DALI improves specification quality, thereby facilitating model reuse in the learnware system.

**Claims And Evidence:**

The claims made in the submission are supported by evidence.

**Essential References Not Discussed:**

No, essential prior works are appropriately cited and discussed.

**Experimental Designs Or Analyses:**

I have generally checked the experimental design and analysis, and they appear sound.

**Methods And Evaluation Criteria:**

Yes, the proposed method and the evaluation criteria make sense for the problem at hand.

**Other Comments Or Suggestions:**

I have no other suggestions.

**Other Strengths And Weaknesses:**

Strengths:
1. This paper proposes a new specification that incorporates the model's discriminative performance.
2. The experiments compare the precision of model search and analyze the privacy protection of the new specification.

Weaknesses:
1. The theoretical analysis (Propositions 4.3-4.6) in this paper is unrelated to the subsequent model search and reuse, and it does not theoretically explain how this specification helps with model search and reuse.
2. DALI improves the consistency of model performance over $R$ and $\mathcal{D}$ by optimizing $\mathcal{L}_{dis}$ during specification generation (which is one of the main contributions of the paper). However, the model search process does not explicitly leverage this characteristic and still primarily relies on a feature distribution-based matching approach.
3. Assuming the number of existing models in the system is $c$, the complexity of solving Eq. (11) should be at least $O(c^2)$. This implies that the proposed model search algorithm has a time complexity that scales at least quadratically with the number of models, whereas existing model search algorithms have a linear time complexity, indicating that the proposed algorithm is more computationally expensive.
4. In the experiments, the performance of DALI shows only a small improvement compared to the contrast method, RKME-W.

**Questions For Authors:**

1. How does DALI compare to RKME in terms of specification generation efficiency?
2. What are the benefits of obtaining $c'$ candidate useful learnwares during the model search process?

**Relation To Broader Scientific Literature:**

The paper contributes to the field of the learnware paradigm by introducing a new learnware specification.

**Theoretical Claims:**

I have generally checked the proofs, but some details have not been thoroughly verified.

---

> ### Author Rebuttal · Authors · 2025-03-31
>
> Thanks for your detailed feedback, and we hope our responses will address your concerns.
>
> Q1: The theoretical analysis is unrelated to the subsequent model search and reuse.
>
> Ans: Our theoretical analysis is closely related to subsequent model search and reuse. Notably, the quality of the generated specification is crucial in the learnware paradigm (please refer to “Introduction” section), as it directly influences the accuracy of subsequent model search and reuse to a certain extent. From the perspectives of loss upper bound (Propositions 4.3 and 4.4), optimization (Proposition 4.5), and privacy protection (Proposition 4.6), our theoretical analysis demonstrates that the proposed \textsc{Dali} method can generate superior specifications, thereby indirectly providing theoretical support for subsequent model search and reuse to some extent.
>
> Q2: The model search process still relies on a feature distribution-based matching approach.
>
> Ans: In the model search process, we perform class feature distribution-based matching  approach on the generated specifications (Eq.(11)). Notably, the specifications are optimized by Eq.(4), and their generation process has fully considered the distribution properties and discriminative performance of the model. Therefore, the matching approach in Eq.(11) does not overlook these characteristics of the model.
>
> Q3: Questions on the efficiency of model search.
>
> Ans: The model search process of our \textsc{Dali} method is comparable in time complexity to existing methods such as RKME, RKME-W, and LANE, all of which require solving a quadratic programming problem similar to Eq.(11) with a complexity of at least $\mathcal{O}(c^{3})$. This is because the submitted requirement task may not find a single specification in the learnware dock system that can fully solve the task but may need to combine multiple specifications to accomplish it. To the best of our knowledge, efficiently conducting model search remains an open problem in learnware research, requiring further in-depth exploration in the future.
>
> Q4: Small improvement compared to RKME-W.
>
> Ans: RKME-W is an extension of the RKME method, incorporating knowledge distillation to retain network parameters as part of the specification. As a result, this method has a significantly larger resource overhead. In contrast, our proposed method outperforms RKME-W while maintaining a lower resource overhead, which we consider a significant improvement. We will emphasize this point in the revised version.
>
> Q5: Comparison of specification generation efficiency with RKME.
>
> Ans: We empirically evaluated the specification generation time of the \textsc{Dali} and RKME methods on datasets under the homogeneous label space setting. The average runtime of \textsc{Dali} was 22.51s, while RKME required 14.05s. Since our approach incorporates random neural embedding mappings, its generation time is indeed slightly longer than RKME’s kernel mapping in this scenario. However, the difference remains within an acceptable range. Notably, \textsc{Dali} achieves a significant performance improvement over RKME at a lower computational cost, a point we will emphasize in the revised version.
>
> Q6: The benefits of obtaining $c'$ candidate useful learnwares.
>
> Ans: One important purpose of learnware paradigm is to enable well-trained models in the dock system to be used "beyond the capabilities of any single model". That is to say, in the learnware dock system, the user can get more than one learnware to solve the user's requirement. For example, the dock system contains learnware related to cucumber, tomato, orange, apple, cabbages, and the user's requirement task is fruit. Then the useful learnwares (candidate useful learnwares) for the requirement may be orange, tomato, and apple. Then, we don't know which part of the task corresponds to orange or apple. At this point, we need further obtain the relationship between the task and the useful learnwares (candidate useful learnwares) to enable model reuse.

---

> > ### Comment · Reviewer_PFro · 2025-04-05
> >
> > Thank you to the authors for the responses and clarifications. Part of my questions have been addressed. I have adjusted my score accordingly.

---

> > > ### Author Response · Authors · 2025-04-06
> > >
> > > Dear Reviewer PFro:
> > >
> > > Thank you so much for your kind reply and for adjusting the score! If you have any questions, don't hesitate to let us know, and we'll do our best to address your concerns.
> > >
> > > Best
> > >
> > > Authors

---

### Decision · Program_Chairs · 2025-05-01

**Decision:**

Accept (poster)

**Comment:**

## Summary
This paper addresses the model specification problem in the learnware paradigm, where users select pre-trained models based on their specifications instead of training new models. The authors argue that existing specification methods based on distribution alignment do not fully reflect the model’s classification ability. To address this, they propose a method called Dali (Dual Alignment), which combines distribution alignment with a second component called discriminative alignment. The goal is to capture a model’s strengths for better reuse. The paper presents theoretical and empirical results showing that this method improves model selection performance.

## Evaluation and Decision

This paper introduces a new learnware paradigm that is useful when handling well-trained models. The reviewers have raised several important concerns and provided feedback to the authors. The authors have done a very good job addressing those concerns. Thus, with the interesting contributions of this paper, I recommend it for acceptance.  The proposed paradigm is interesting, and the specification design is quite an important problem. The paper also provides a theoretical analysis of the proposed specification. It proposes a new specification that incorporates the model's discriminative performance. It makes a significant contribution to the learnware paradigm. By bringing in a new perspective, it enhances the overall capability of the learnware system and could serve as valuable inspiration for future research. The experiments compare the precision of model search and analyze the privacy protection of the new specification.